# The molecular mechanism and evolutionary divergence of caspase 3/7-regulated gasdermin E activation

Hang Xu[1,2,3], Zihao Yuan[1,2], Kunpeng Qin[1,2,3], Shuai Jiang[1,2,3]*, Li Sun[1,2,3]*

[1]CAS and Shandong Province Key Laboratory of Experimental Marine Biology, Institute of Oceanology; CAS Center for Ocean Mega-Science, Chinese Academy of Sciences, Qingdao, China; [2]Laboratory for Marine Biology and Biotechnology, Qingdao Marine Science and Technology Center, Qingdao, China; [3]College of Marine Sciences, University of Chinese Academy of Sciences, Qingdao, China

**Abstract** Caspase (CASP) is a family of proteases involved in cleavage and activation of gasdermin, the executor of pyroptosis. In humans, CASP3 and CASP7 recognize the same consensus motif DxxD, which is present in gasdermin E (GSDME). However, human GSDME is cleaved by CASP3 but not by CASP7. The underlying mechanism of this observation is unclear. In this study, we identified a pyroptotic pufferfish GSDME that was cleaved by both pufferfish CASP3/7 and human CASP3/7. Domain swapping between pufferfish and human CASP and GSDME showed that the GSDME C-terminus and the CASP7 p10 subunit determined the cleavability of GSDME by CASP7. p10 contains a key residue that governs CASP7 substrate discrimination. This key residue is highly conserved in vertebrate CASP3 and in most vertebrate (except mammalian) CASP7. In mammals, the key residue is conserved in non-primates (e.g., mouse) but not in primates. However, mouse CASP7 cleaved human GSDME but not mouse GSDME. These findings revealed the molecular mechanism of CASP7 substrate discrimination and the divergence of CASP3/7-mediated GSDME activation in vertebrate. These results also suggested that mutation-mediated functional alteration of CASP probably enabled the divergence and specialization of different CASP members in the regulation of complex cellular activities in mammals.

**\*For correspondence:**
sjiang@qdio.ac.cn (SJ);
lsun@qdio.ac.cn (LS)

**Competing interest:** The authors declare that no competing interests exist.

## eLife assessment

This **important** study elucidates the molecular divergence of caspase 3 and 7 in the vertebrate lineage. **Convincing** biochemical and mutational data provide evidence that in humans, caspase 7 has lost the ability to cleave gasdermin E due to changes in a key residue, S234. The diversification and specialization of gasdermins such as gasdermin E in humans compared to early vertebrates such as teleosts may enable each human gasdermin molecule to have more restricted and tightly regulated physiological functions in different cell death pathways.

## Introduction

Pyroptosis represents a form of programmed cell death that provokes robust inflammatory immune response (*Bergsbaken et al., 2009*; *Tsuchiya et al., 2019*). Gasdermin (GSDM) serves as the direct executioner of pyroptosis. GSDM forms transmembrane pores to permeabilize the cytoplasmic membrane, which leads to the release of pro-inflammatory cytokines and, if these transmembrane pores persist, pyroptosis (*Kovacs and Miao, 2017*; *Shi et al., 2017*; *Xia et al., 2021*). Humans have six GSDM (HsGSDM) family members, HsGSDMA-E, and HsPJVK (*Tamura et al., 2007*). All HsGSDMs

**eLife digest** Cell death is essential for an organism to develop and survive as it plays key roles in processes such as embryo development and tissue regeneration. Cell death is also an important form of defence during an infection. A form of programmed cell death known as pyroptosis can be induced in infected cells, which helps to kill the infectious agent as well as alert the immune system to the infection.

Pyroptosis is driven by Gasdermin E, a protein made up of two domains. At one end of the protein, the 'N-terminal' domain punctures holes in cell membranes, which can lead to cell death. At the other end, the 'C-terminal' domain inhibits the activity of the N-terminal domain. A family of proteins called caspases activate Gasdermin E by cleaving it, which releases the N-terminal domain from the inhibitory C-terminal domain. In humans, two caspases known as CASP3 and CASP7 recognize a specific sequence of amino acids – the building blocks of proteins – in Gasdermin E. However, only CASP3 is able to cleave the protein.

After discovering that, unlike in humans, pufferfish Gasdermin E can be cleaved by both CASP3 and CASP7, Xu et al. wanted to investigate the underlying mechanisms behind this difference. Swapping the domains of human and pufferfish Gasdermin E and creating different versions of CASP7 revealed that the C-terminal domain of Gasdermin E and a single amino acid in CASP7 determine whether cleavage is possible. Interestingly, the key amino acid sequence required for cleavage by CASP7 is present in most vertebrate CASP3 and CASP7 proteins. However, it is absent in most mammalian CASP7.

The findings of Xu et al. suggest that the different activity of human CASP7 and CASP3 is driven by a single amino acid mutation. This change likely played an important role in the process of different CASP proteins evolving to regulate different cellular activities in mammalian cells. This knowledge will be useful for future studies on the evolution and specialization of other closely related proteins.

(except for HsPJVK) adopt a two-domain architecture, the N-terminal (NT) pore-forming domain and the C-terminal (CT) autoinhibitory domain (*De Schutter et al., 2021*; *Shao, 2021*). Proteolytic cleavage of HsGSDM to remove the autoinhibitory CT domain enables the binding of the lipophilic NT fragment to the cell membrane, where the NT fragments oligomerize and form transmembrane pores to induce osmotic cell lysis (*Kuang et al., 2017*; *Liu et al., 2016*).

Among the signaling pathways that activate GSDM-mediated pyroptosis, caspase (CASP) 1/4/5/11-mediated GSDMD activation is well documented. In humans, HsGSDMD is specifically cleaved by the inflammatory HsCASP1 through multiple inflammasome signaling pathways. HsCASP4/5 also cleave HsGSDMD in response to lipopolysaccharide (LPS) stimulation. After cleavage by HsCASP1/4/5, the pore-forming HsGSDMD-NT fragment becomes unconstrained and triggers pyroptosis (*He et al., 2015*; *Kayagaki et al., 2015*; *Shi et al., 2015*). Structural and biochemical analyses reveal that HsCASP1/4 βIII/βIII' sheet forms an exosite that interacts with the hydrophobic pocket of HsGSDMD-CT domain, rendering HsGSDMD cleavage independent of CASP recognition of the tetrapeptide motif FLTD (*Wang et al., 2020b*). When the tetrapeptide FLTD is mutated to AAAD, HsCASP1/4 cleavage of HsGSDMD is not affected. The residues that form the hydrophobic pocket of HsGSDMD-CT domain are not conserved in HsGSDME, suggesting that HsGSDME cannot be cleaved by HsCASP1/4 via interaction with an exosite (*Liu et al., 2020*). Actually, human HsGSDME is specifically cleaved by HsCASP3 at the consensus motif DMPD, which is required for HsCASP3-mediated HsGSDME proteolysis. Mutation of either the Asp residue of the DMPD tetrapeptide motif leads to cleavage resistance of HsGSDME (*Rogers et al., 2017*; *Wang et al., 2017*), indicating that HsCASP3-mediated HsGSDME activation paradigm is distinct from that of HsGSDMD. It is intriguing that although HsCASP7 shares the same recognition motif (DxxD) with HsCASP3, HsCASP7 cannot cleave HsGSDME (*Agniswamy et al., 2009*; *Slee et al., 2001*). The molecular mechanism underlying the HsCASP3/7 cleavage discrimination of HsGSDME is unknown.

Historically, HsCASP3 and HsCASP7, which are 54% identical, are known to possess similar structure and share overlapping protein substrate repertoire, and are considered to have functionally redundant roles in regulating cell death (*Crawford and Wells, 2011*; *Kumar, 2007*; *Shi, 2002*). Both HsCASP3 and HsCASP7 consist of two polypeptide subunits named p20 and p10 (1:1 ratio), which assemble

to form active heterotetramer (*Timmer and Salvesen, 2007*). HsCASP3/7 contain highly conserved QACRG and SHG motifs in p20 subunit, and SWR and GSWF motifs in p10 subunit, which participate in the catalytic reaction and substrate binding, respectively (*Boyce et al., 2004*; *Cohen, 1997*). In the apoptosis signaling pathway, death receptor-mediated extrinsic pathway and mitochondria-mediated intrinsic pathway ultimately converge to the activation of HsCASP3/7 (*Budd et al., 2000*; *Wajant, 2002*). Active HsCASP3/7 cleave a series of protein substrates, including poly(ADP-ribose) polymerase and DNA fragmentation factor 45, which causes chromatin fragmentation and leads to apoptosis (*Erener et al., 2012*; *Wolf et al., 1999*; *Zheng et al., 1998*). Although HsCASP3 and HsCASP7 exert almost indistinguishable proteolytic specificity to certain polypeptides, there exist functional differences between these two HsCASPs in cleaving some substrates (*Brentnall et al., 2013*; *Demon et al., 2009*). For instance, HsCASP3 cleaves Bid much more efficiently than HsCASP7, while HsCASP7 cleaves cochaperone p23 more efficiently than HsCASP3 (*Slee et al., 2001*; *Walsh et al., 2008*). These findings support the notion that HsCASP3 and HsCASP7 are enzymatically similar but functionally non-redundant.

Different from HsGSDME that is specifically cleaved by HsCASP3, teleost GSDME is cleaved via various modes (*Angosto-Bazarra et al., 2022*; *Yuan et al., 2022*). In this study, we identified a functional GSDME (named TrGSDME) from pufferfish (*Takifugu rubripes*). We found that TrGSDME was specifically cleaved by both HsCASP3/7 and TrCASP3/7, whereas HsGSDME was cleaved by TrCASP3/7 and HsCASP3, but not by HsCASP7. We examined the underlying mechanism of this observation. We found that the GSDME-CT and the CASP7 p10 were critical for CASP7 cleavage of GSDME. By a series of site-directed mutagenesis, we discovered a previously unrecognized key residue (S234 in HsCASP7) in p10 that was responsible for the discriminative cleavage of HsGSDME by HsCASP7.

## Results

### Pufferfish GSDME is specifically cleaved by both human and fish CASP3/7

It has long been observed that although human caspase (HsCASP) 3 and 7 share the same consensus recognition motif DxxD, HsCASP3, but not HsCASP7, cleaves human GSDME (HsGSDME) (at the site of DMPD) (*Poreba et al., 2013*; *Wang et al., 2017*). The underlying molecular mechanism is unknown. In this study, we found that a pufferfish *T. rubripes* GSDME (designated TrGSDME), which belongs to the GSDMEa lineage of teleost GSDME (*Figure 1—figure supplement 1*), was specifically cleaved by both HsCASP3 and 7 (*Figure 1A and B*). This intriguing observation promoted us to explore the mechanism of HsCASP7 substrate discrimination. We first examined whether TrGSDME could be cleaved by pufferfish CASP (TrCASP) 3/7. The active forms of TrCASP3/7 were prepared (*Figure 1C*), both exhibited high proteolytic specificity and activity toward the tetrapeptide DEVD (*Figure 1D and E*), which is the conserved recognition motif of HsCASP3/7. When incubated with TrGSDME, both TrCASP3 and TrCASP7 cleaved TrGSDME into the NT and CT fragments, similar to that cleaved by HsCASP3/7, in a dose-dependent manner (*Figure 1F and G*). Accordingly, TrCASP3/7-mediated TrGSDME cleavage was inhibited by the CASP3 inhibitor (Z-DEVD-FMK) and the pan-CASP inhibitor (Z-VAD-FMK) (*Figure 1H*). Based on the molecular mass of the cleaved NT and CT products, we inferred that the tetrapeptide motif $^{255}$DAVD$^{258}$ in the vicinity of the linker region of TrGSDME may be the recognition site of CASP3/7. Indeed, the D255R and D258A mutants of TrGSDME were resistant to TrCASP3/7 and HsCASP3/7 cleavage (*Figure 1I and J*, *Figure 1—figure supplement 2*). Taken together, these results demonstrated that TrGSDME was cleaved by fish and human CASP3/7 in a manner that depended on the specific sequence of DAVD (*Figure 1K*).

### CASP3/7-cleaved TrGSDME is functionally activated and induces pyroptosis

We next examined whether TrGSDME, like HsGSDME, is a functional pyroptosis inducer. For this purpose, HEK293T cells were transfected with mCherry-tagged full length (FL) or NT/CT domain of TrGSDME. The results revealed that TrGSDME-FL and -CT were abundantly expressed in the cells, whereas TrGSDME-NT expression was barely detectable (*Figure 2A*, *Figure 2—figure supplement 1*). No significant morphological change or lactate dehydrogenase (LDH) release was observed in cells expressing TrGSDME-FL or -CT (*Figure 2B and C*). By contrast, cells expressing TrGSDME-NT showed

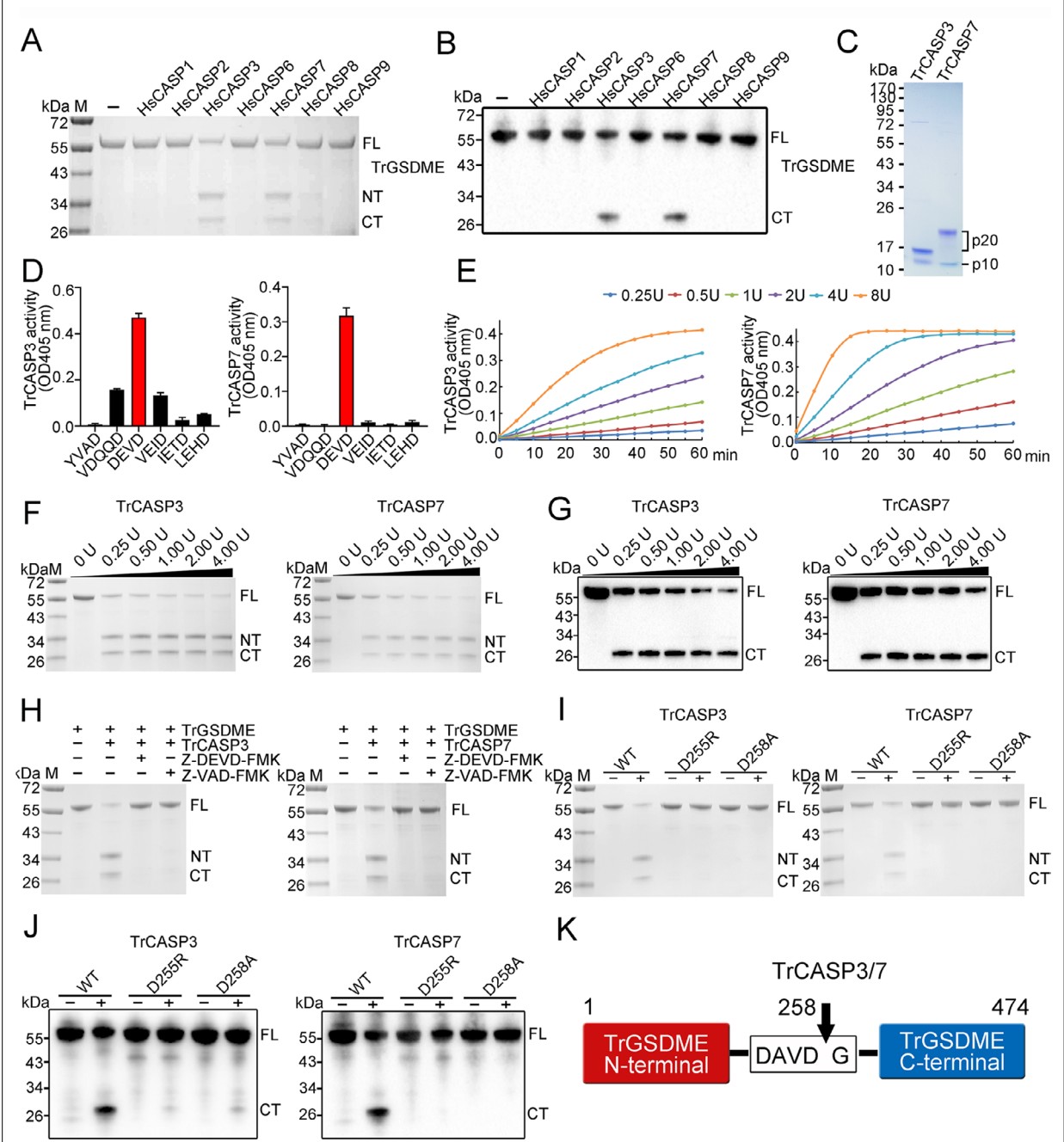

**Figure 1.** Cleavage of TrGSDME by caspases. (**A, B**) TrGSDME was treated with 1 U of different HsCASPs for 2 hr and then subjected to SDS-PAGE (**A**) and immunoblotting (**B**). (**C**) SDS-PAGE analysis of purified TrCASP3/7. The p10 and p20 subunits are indicated. (**D**) TrCASP3/7 cleavage of different colorimetric CASP substrates was monitored by measuring released $\rho$NA. n=3. The values are the means ± SD of three replicates. (**E**) TrCASP3/7 (0.25–8 U) were incubated with Ac-DEVD-$\rho$NA, and time-dependent release of $\rho$NA was measured. (**F, G**) SDS-PAGE (**F**) and immunoblotting (**G**) analysis of TrGSDME cleavage by TrCASP3/7 (0.25–4 U). (**H**) TrGSDME cleavage by TrCASP3/7 was determined in the presence or absence of 20 μM Z-DEVD-FMK or Z-VAD-FMK. (**I, J**) TrGSDME wild type (WT) and mutants (D255R or D258A) were incubated with TrCASP3/7 for 2 hr, and the cleavage was determined by SDS-PAGE (**I**) and immunoblotting (**J**). (**K**) A schematic of TrGSDME cleavage by TrCASP3/7. The arrow indicates cleavage site. For all panels, FL, full length; NT, N-terminal fragment; CT, C-terminal fragment.

The online version of this article includes the following video, source data, and figure supplement(s) for figure 1:

**Source data 1.** Original file for the western blot or SDS-PAGE analysis in *Figure 1*.

**Source data 2.** File containing *Figure 1A–C and F–J* and original scans of the relevant western blot or SDS-PAGE analysis with highlighted bands.

**Source data 3.** Numerical data from all bar graphs shown in *Figure 1*.

*Figure 1 continued on next page*

*Figure 1 continued*

**Figure supplement 1.** Phylogenetic analysis of the TrGSDME used in this study.

**Figure supplement 2.** Cleavage of TrGSDME mutants by HsCASP3/7.

**Figure supplement 2—source data 1.** Original file for the western blot analysis in *Figure 1—figure supplement 2*.

**Figure supplement 2—source data 2.** File containing *Figure 1—figure supplement 2* and original scans of the relevant western blot analysis with highlighted bands.

**Figure 1—video 1.** TrGSDME-NT induces pyroptosis of HEK293T cells.

https://elifesciences.org/articles/89974/figures#fig1video1

necrotic cell death with massive LDH release and positive Sytox Green staining (*Figure 2B and C*, *Figure 2—figure supplement 2*). TrGSDME-NT-induced cell death exhibited osmotic cell membrane swelling, a typical feature of pyroptosis (*Figure 2D*). Time-lapse imaging showed that TrGSDME-NT triggered rapid cell swelling and membrane rupture, which led to release of cytoplasmic contents and eventually cell death (*Figure 2E*, *Figure 1—video 1*). To examine the pyroptotic activity of CASP3/7-cleaved TrGSDME, TrCASP3/7 and TrGSDME were overexpressed in HEK293T cells (*Figure 2*). The cells expressing TrCASP3, TrCASP7, or TrGSDME alone had no apparent morphological change or LDH release, whereas the cells co-expressing TrGSDME and TrCASP3 or TrGSDME and TrCASP7 underwent pyroptosis, accompanying with massive LDH release, membrane rupture, and TrGSDME cleavage (*Figure 2G–I*, *Figure 2—figure supplement 3A*). Consistently, the presence of the CASP3 inhibitor and pan-CASP inhibitor hampered TrGSDME-mediated pyroptosis (*Figure 2J and K*). Mutation of the cleavage site (D255R and D258A) inhibited pyroptosis and significantly decreased LDH release and TrGSDME cleavage (*Figure 2L and M*, *Figure 2—figure supplement 3B and C*). These results indicated that TrCASP3/7 cleavage was required to activate TrGSDME-mediated pyroptosis.

## GSDME-CT domain mediates the recognition by CASP7

Since it is known that human GSDME is cleaved by CASP3 but not by CASP7, the observation that TrGSDME was cleaved by both human and fish CASP3/7 intrigued us to explore the underlying mechanism. We found that unlike TrCASP1, both HsCASP3 and 7 exhibited proteolytic activity toward the consensus CASP3/7 recognition motif, DMPD, in HsGSDME, but HsCASP7 failed to cleave HsGSDME (*Figure 3A and B*, *Figure 3—figure supplements 1 and 2*). Structure analysis revealed that compared to TrGSDME, HsGSDME possesses two additional regions with the possibility to form a loop (261–266 aa) and an α-helix (281–296 aa), respectively (*Figure 3C*). We tested whether deletion of these two regions could confer HsCASP7 cleavage on HsGSDME. The results showed that similar to the wild type HsGSDME, the loop deletion mutant (Δ261–266) and the α-helix deletion mutant (Δ281–296) were cleaved by HsCASP3 but not by HsCASP7 (*Figure 3D*). Since the NT and CT domains play different roles in GSDM structure maintenance, we constructed GSDME chimeras consisting of HsGSDME-NT plus TrGSDME-CT (named HsNT-TrCT), or TrGSDME-NT plus HsGSDME-CT (named TrNT-HsCT) (*Figure 3E*). Compared with wild type HsGSDME and TrGSDME, chimeric HsNT-TrCT was cleaved by HsCASP3/7, whereas TrNT-HsCT was cleaved only by HsCASP3 (*Figure 3F–I*), suggesting that the CT domain determined the cleavability of GSDME by HsCASP7.

## The p10 subunit determines the substrate specificity of CASP7

Since as shown above, unlike HsCASP7, TrCASP7 was able to cleave HsGSDME (*Figure 3H*), we compared the sequences of TrCASP7 and HsCASP7. The two CASPs share 66.24% sequence identity. In these CASPs, the catalytic motifs SHG and QACRG in the p20 subunit and the substrate binding motifs SWR and GSWF in the p10 subunit are conserved (*Figure 4A*). To explore their substrate discrimination mechanism, we constructed CASP7 chimeras consisting of HsCASP7 p20 plus TrCASP7 p10 (named Hsp20-Trp10) or TrCASP7 p20 plus HsCASP7 p10 (named Trp20-Hsp10) (*Figure 4B*, *Figure 4—figure supplement 1A*). Compared to the wild types (HsCASP7 and TrCASP7), the chimeras exhibited comparable enzymatic activities toward the tetrapeptide substrates DAVD and DMPD (*Figure 4C*). Like the wild types, both chimeras cleaved TrGSDME and HsNT-TrCT (*Figure 4D and E*). By contrast, the Hsp20-Trp10 chimera cleaved HsGSDME, whereas the Trp20-Hsp10 chimera did not cleave HsGSDME (*Figure 4F*). Similar cleavage pattern was observed toward TrNT-HsCT

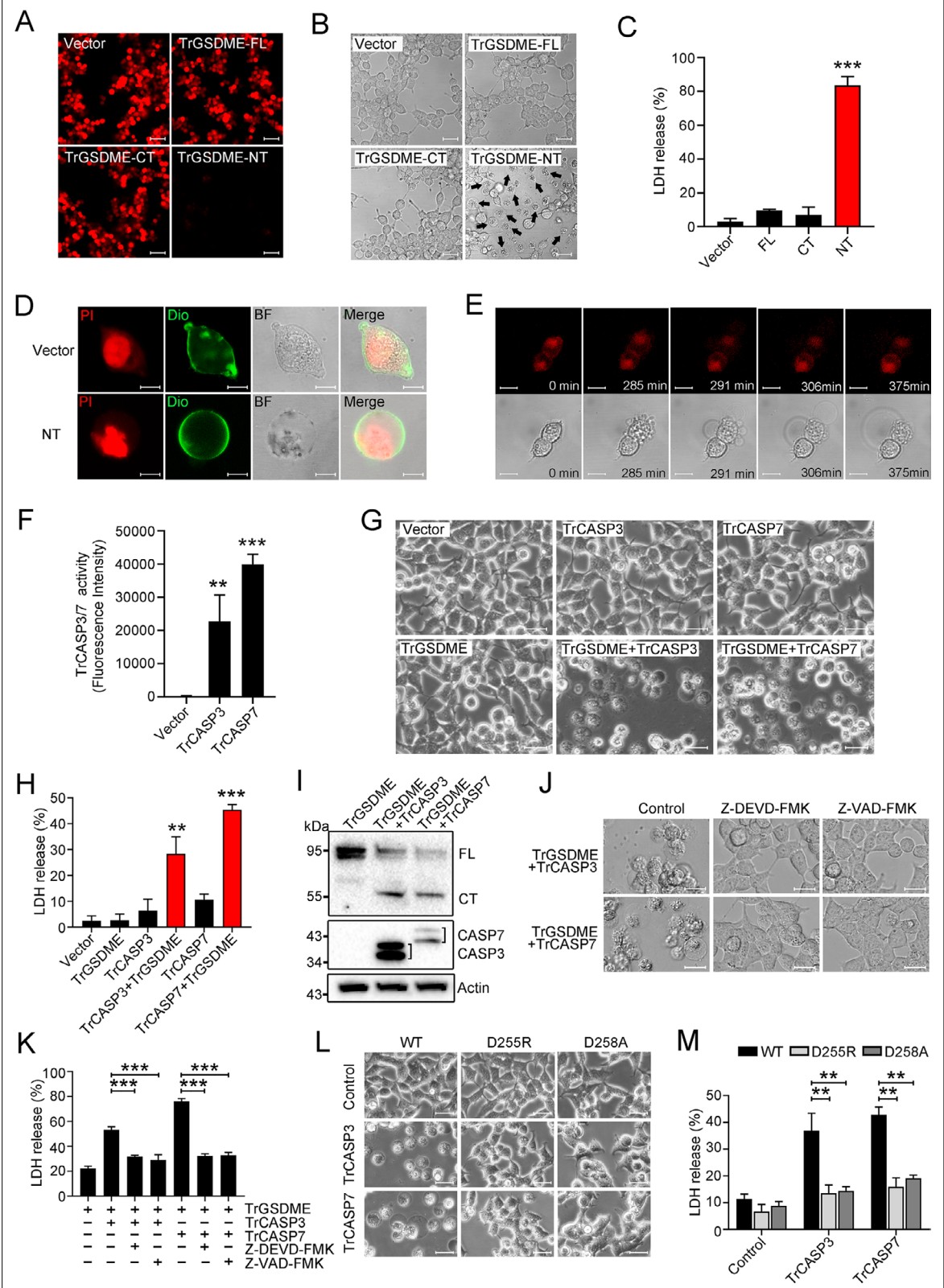

**Figure 2.** The pyroptotic activity of TrGSDME. (**A–C**) HEK293T cells were transfected with C-terminally mCherry-tagged TrGSDME FL or truncate (NT or CT) for 24 hr, and then analyzed for TrGSDME expression (**A**), morphological change (**B**), and lactate dehydrogenase (LDH) release (**C**). Scale bars, 50 μm (**A**) and 20 μm (**B**). (**D**) HEK293T cells were transfected with the backbone vector or vector expressing Myc-tagged TrGSDME-NT for 24 hr. The cell nuclei and membrane were stained with propidium iodide (PI) and DiO, respectively. Scale bar, 10 μm. (**E**) HEK293T cells were transfected with

*Figure 2 continued on next page*

*Figure 2 continued*

mCherry-tagged TrGSDME-NT for 16 hr, and the progression of cell death was shown by time-lapse microscopic imaging. Scale bar, 10 μm. (**F**) HEK293T cells were transfected with TrCASP3/7 for 24 hr, and the proteolytic activity of TrCASP3/7 was assessed by treatment with fluorogenic Ac-DEVD-AFC. (**G**) Phase-contrast images of HEK293T cells transfected with the indicated vectors for 24 hr. Scale bar, 20 μm. (**H, I**) LDH release (**H**) and TrGSDME cleavage and TrCASP3/7 expression (**I**) in the above transfected cells were determined. For panel (**I**), FL, full-length; CT, C-terminal fragment. (**J, K**) HEK293T cells co-expressing TrGSDME and TrCASP3/7 for 24 hr in the presence or absence (control) of 10 μM Z-VAD-FMK or Z-DEVD-FMK were subjected to microscopy (**J**) and LDH measurement (**K**). Scale bar, 20 μm. (**L, M**) HEK293T cells expressing TrCASP3/7 plus TrGSDME or TrCASP3/7 plus the D255R/D258A mutant were subjected to microscopy (**L**) and LDH measurement (**M**). Scale bar, 20 μm. For panels (**C, F, H, K, M**), values are the means of three experimental replicates and shown as means ± SD. n=3. **p<0.01, ***p<0.001.

The online version of this article includes the following source data and figure supplement(s) for figure 2:

**Source data 1.** Original file for the western blot analysis in *Figure 2*.

**Source data 2.** File containing *Figure 2I* and original scans of the relevant western blot analysis with highlighted bands.

**Source data 3.** Numerical data from all bar graphs shown in *Figure 2*.

**Figure supplement 1.** Ectopic expression of TrGSDME-FL and truncates.

**Figure supplement 1—source data 1.** Original file for the western blot analysis in *Figure 2—figure supplement 1*.

**Figure supplement 1—source data 2.** File containing *Figure 2—figure supplement 1* and original scans of the relevant western blot analysis with highlighted bands.

**Figure supplement 2.** The cell death-inducing ability of TrGSDME-FL and truncates.

**Figure supplement 3.** Activation of TrGSDME-mediated pyroptosis by TrCASP3/7.

**Figure supplement 3—source data 1.** Original file for the western blot analysis in *Figure 2—figure supplement 3*.

**Figure supplement 3—source data 2.** File containing *Figure 2—figure supplement 3C* and original scans of the relevant western blot analysis with highlighted bands.

(*Figure 4G*). These observations indicated that the p10 subunit dictated the cleavage specificity of CASP7 toward GSDME.

## The S234 of p10 is the key to HsCASP7 discrimination on HsGSDME

Comparative analysis of the p10 sequences of TrCASP7 and HsCASP7 identified 13 non-conserved residues. To examine their potential involvement in GSDME cleavage, a series of site-directed mutagenesis was performed to swap the non-conserved residues between HsCASP7 p10 and TrCASP7 p10 (*Figure 5A*, *Figure 4—figure supplement 1B and C*). Of the 13 swaps created, S234N conferred on HsCASP7 the ability to cleave HsGSDME (*Figure 5B*), whereas its corresponding swap in TrCASP7 (N245S) markedly reduced the ability of TrCASP7 to cleave HsGSDME (*Figure 5C*). None of the 13 swaps affected the ability of HsCASP7 or TrCASP7 to cleave TrGSDME (*Figure 5D and E*). HsCASP7-S234N cleaved HsGSDME in a dose-dependent manner (*Figure 5F*), and the cleavage was not enhanced by the additional mutation of S247N&I248V (*Figure 5G*). Additionally, HsCASP7 and HsCASP7-S234N exhibited similar cleavage capacities against other CASP7 substrates, such as PARP1 and gelsolin (*Figure 5—figure supplement 1*, *Erener et al., 2012*; *Walsh et al., 2008*). Previous studies showed that human CASP1/4 cleaved GSDMD through exosite interaction (*Liu et al., 2020*; *Wang et al., 2020b*). However, this exosite is not conserved in HsCASP7, in which the corresponding region forms a coil, not a β sheet (*Figure 5—figure supplement 1*). Structural modeling showed that the residues Q276, D278, and H283 are close to the region corresponding to the HsCASP1/4 exosite (*Figure 5—figure supplement 1*). Since these three residues are also not conserved between HsCASP7 p10 subunit and TrCASP7 p10 subunit, they were selected for mutagenesis to examine whether there existed an exosite interaction between HsCASP7 and HsGSDME. The mutation results showed that, similar to the wild type, the mutant Q276W&D278E&H283S was unable to cleave HsGSDME (*Figure 5G*), suggesting that, unlike human GSDMD, HsGSDME cleavage by CASPs probably did not involve exosite interaction. In contrast to HsGSDME, TrGSDME cleavage by HsCASP7 was not affected by the mutation of S234N, S234N plus S247N&I248V, or Q276W&D278E&H283S (*Figure 5H*).

All the above observed GSDME cleavages by CASP7 were verified in a cellular system. When co-expressed in HEK293T cells, HsGSDME was cleaved by TrCASP7 but not by HsCASP7, while TrGSDME was cleaved by both TrCASP7 and HsCASP7 (*Figure 5—figure supplement 3*). Additionally, both HsGSDME and TrGSDME were cleaved by HsCASP7-S234N as well as TrCASP7-N245S (*Figure 5I*

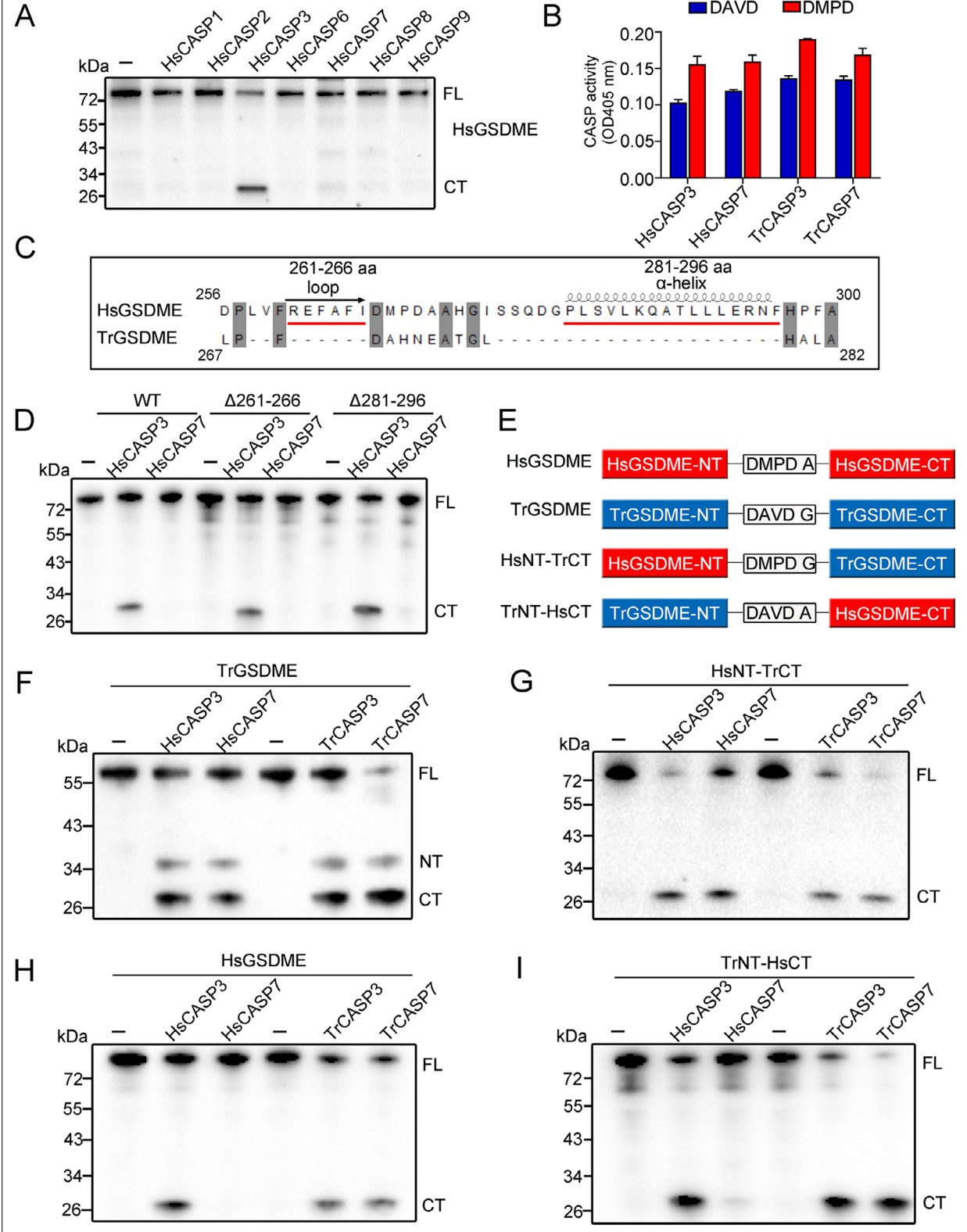

**Figure 3.** GSDME-CT domain determines the recognition by CASP7. (**A**) HsGSDME was treated with 1 U of various HsCASPs for 2 hr, and the products were analyzed by immunoblotting with anti-HsGSDME-CT antibody. (**B**) HsCASP3/7 and TrCASP3/7 were incubated with Ac-DAVD-$\rho$NA or Ac-DMPD-$\rho$NA at 37°C for 30 min, and the proteolytic activity was determined. n=3. Data are expressed as the means ± SD of three replicates. (**C**) Structure analysis of the linker region from HsGSDME and TrGSDME with moue GSDMA3 (PDB:5b5r) as the template. Two regions that may form α-helix and

*Figure 3 continued on next page*

*Figure 3 continued*

loop are indicated. (**D**) HsGSDME wild type (WT) and mutants (Δ261–266 and Δ281–296) were treated with or without 1 U of HsCASP3/7 for 2 hr, and the cleaved fragments were subjected to immunoblotting with anti-HsGSDME-CT antibody. (**E**) Schematics of the chimeric GSDME constructs. (**F–I**) TrGSDME (**F**), HsNT-TrCT (**G**), HsGSDME (**H**), and TrNT-HsCT (**I**) were incubated with 1 U of TrCASP3/7 or HsCASP3/7 for 2 hr, and the cleavage was determined by immunoblotting with anti-TrGSDME or anti-HsGSDME-CT antibodies. For panels (**A, D, F–I**): FL, full length; NT, N-terminal fragment; CT, C-terminal fragment.

The online version of this article includes the following source data and figure supplement(s) for figure 3:

**Source data 1.** Original file for the western blot analysis in *Figure 3*.

**Source data 2.** File containing *Figure 3A, D–F, and I* and original scans of the relevant western blot analysis with highlighted bands.

**Source data 3.** Numerical data from the bar graph shown in *Figure 3*.

**Figure supplement 1.** Cleavage of HsGSDME by caspases.

**Figure supplement 1—source data 1.** Original file for the SDS-PAGE analysis in *Figure 3—figure supplement 1*.

**Figure supplement 1—source data 2.** File containing *Figure 3—figure supplement 1* and original scans of the relevant SDS-PAGE analysis with highlighted bands.

**Figure supplement 2.** Proteolytic activity of TrCASP1.

**Figure supplement 2—source data 1.** Numerical data from the bar graph shown in *Figure 3—figure supplement 2*.

---

*and J*). An apparent dose effect was observed in the cleavage of HsGSDME by HsCASP7-S234N (*Figure 5K*).

## Divergent and GSDME-independent evolution of CASP7 leads to the loss of GSDME-activation function in mammalian CASP7

Given the importance of N234 in p10 for HsCASP7 cleavage of HsGSDME, we analyzed the sequence conservation of this locus in CASP3/7. For CASP3, a highly conserved Asn residue (corresponding to HsCASP7 S234) was found immediately after the SWR motif in vertebrates (*Figure 6A and B*). In HsCASP3, this conserved Asn is at position 208. We examined whether this residue was functionally essential to HsCASP3. Compared to the wild type, the N208S mutant exhibited much weaker cleavage of HsGSDME (*Figure 6C*, *Figure 4—figure supplement 1D*). Similar weakened cleavage was also observed in HEK293T cells co-expressing HsCASP3-N208S and HsGSDME (*Figure 6D*). These results indicated that the Asn residue was also critical for HsCASP3 cleavage of HsGSDME.

For CASP7, an Asn at the position corresponding to HsCASP7 S234 is highly conserved in teleost, amphibians, reptiles, and birds, but not in mammals (*Figure 6A*). In mammals, the corresponding Asn is present in most non-primate species, such as mouse (*Mus musculus*) and bovine (*Bos taurus*), but is replaced by Ser in primate, such as humans (*Homo sapiens*), gorillas (*Gorilla gorilla*), and chimpanzee (*Pan troglodytes*) (*Figure 6E*, *Figure 6—figure supplement 1*). We examined whether mouse CASP7 (MmCASP7) could cleave GSDME. The results showed that MmCASP7 cleaved HsGSDME in a dose-dependent manner, and the cleaving ability was abrogated by N234S mutation (*Figure 6G and H*, *Figure 4—figure supplement 1E*). By contrast, neither MmCASP7 nor MmCASP7-N234S was able to cleave mouse GSDME (MmGSDME) (*Figure 6I*). Mouse CASP3 (MmCASP3), however, cleaved MmGSDME efficiently (*Figure 6—figure supplement 2*). These results indicated the existence of an intra-species barrier for GSDME cleavage by CASP7 in some mammals, suggesting independent evolutions of CASP7 and GSDME.

## Discussion

GSDME is an ancient member of the GSDM family existing ubiquitously in vertebrate from teleost to mammals (*Broz et al., 2020*; *De Schutter et al., 2021*). In humans, GSDME is specifically cleaved by CASP3 at the consensus motif DMPD, but it is not cleaved by CASP7, which recognizes the same DxxD motif. CASP3 cleavage releases the pyroptosis-inducing NT fragment from the association of the inhibitory CT fragment and switches cell death from apoptosis to pyroptosis (*Rogers et al., 2017*; *Wang et al., 2017*). In this study, we found that different from human GSDME, a pufferfish GSDME belonging to the teleost GSDMEa lineage was specifically cleaved by both CASP3 and CASP7 at the site of DAVD to liberate the pyroptosis-inducing NT fragment. In teleost, there generally exist two GSDME orthologs, named GSDMEa and GSDMEb. Comparative study on the functions of GSDMEa

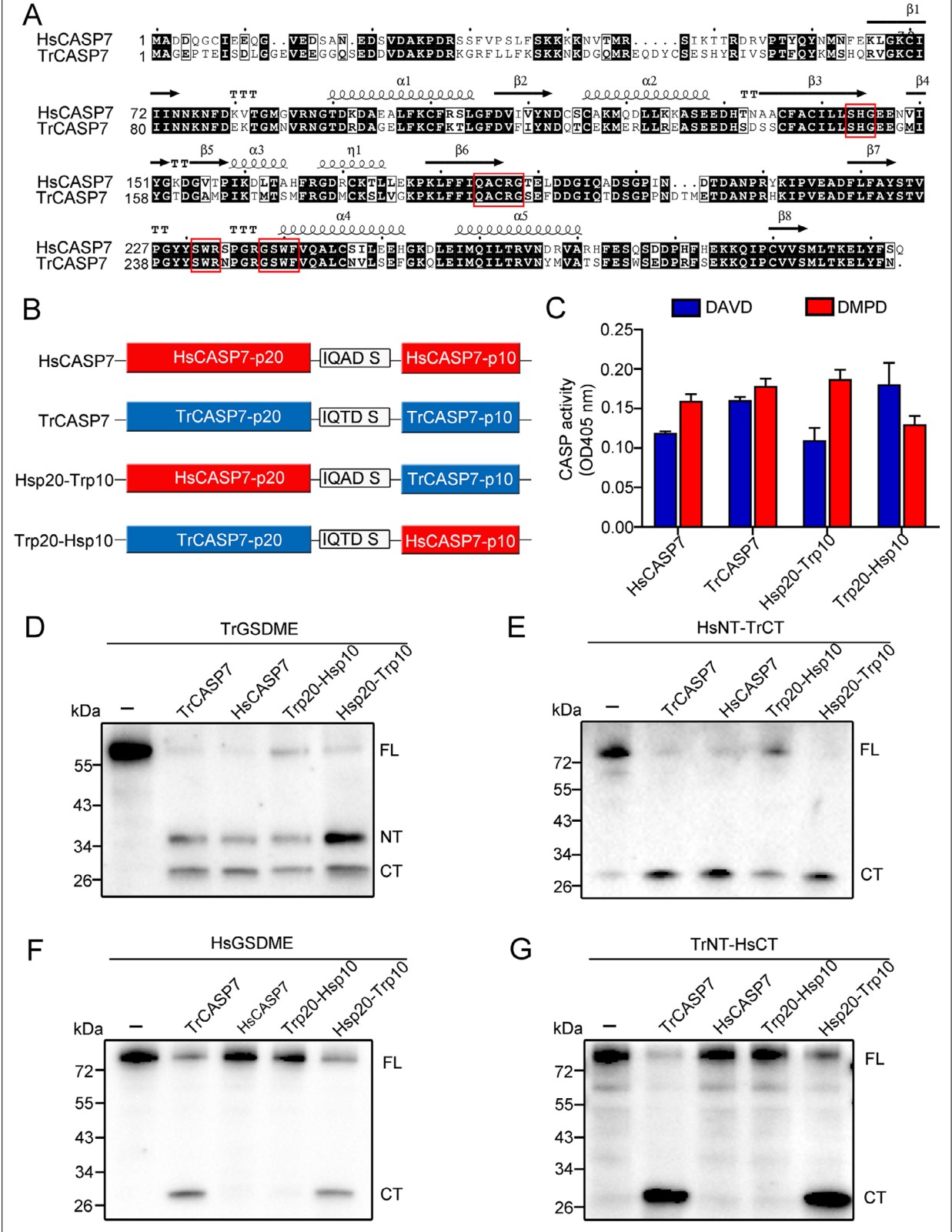

**Figure 4.** The p10 subunit of CASP7 is essential for discrimination on GSDME. (**A**) Sequence alignment of HsCASP7 and TrCASP7 with HsCASP7 (PDB:1K86) as the template. α-helices and β-strands are indicated. The motifs involved in catalytic reaction (SHG and QACRG) and substrate binding (SWR and GSWF) are boxed in red. (**B**) Schematics of CASP7 chimeras. (**C**) The proteolytic activities of TrCASP7, HsCASP7, and their chimeras were determined by cleaving Ac-DAVD-ρNA and Ac-DMPD-ρNA. n=3. Data are expressed as the means ± SD of three replicates. (**D–G**) TrGSDME

*Figure 4 continued on next page*

*Figure 4 continued*

(**D**), HsNT-TrCT (**E**), HsGSDME (**F**), or TrNT-HsCT (**G**) were incubated with TrCASP7, HsCASP7 and their chimeras for 2 hr. The products were assessed by immunoblotting with anti-TrGSDME or anti-HsGSDME-CT antibodies. For panels (**D–G**): FL, full length; NT, N-terminal fragment; CT, C-terminal fragment.

The online version of this article includes the following source data and figure supplement(s) for figure 4:

**Source data 1.** Original file for the western blot analysis in *Figure 4*.

**Source data 2.** File containing *Figure 4D–G* and original scans of the relevant western blot analysis with highlighted bands.

**Source data 3.** Numerical data from the bar graph shown in *Figure 4*.

**Figure supplement 1.** Preparation of recombinant proteins.

**Figure supplement 1—source data 1.** Original file for the western blot or SDS-PAGE analysis in *Figure 4—figure supplement 1*.

**Figure supplement 1—source data 2.** File containing *Figure 4—figure supplement 1A–E* and original scans of the relevant western blot or SDS-PAGE analysis with highlighted bands.

and GSDMEb is scarce, and it remains to be explored whether clear physiological roles are played by these two orthologs in fish. As executors of pyroptosis, teleost GSDMEs are activated via different mechanisms. In tongue sole (*Cynoglossus semilaevis*), GSDMEb is preferably cleaved by CASP1 to trigger pyroptosis (*Jiang et al., 2019*); in zebrafish (*Danio rerio*), GSDMEb is activated by caspy2 (CASP5 homolog) through the NLR family pyrin domain containing 3 (NLRP3) inflammasome signaling pathway (*Li et al., 2020*; *Wang et al., 2020a*). These observations indicate that similar to mammalian GSDMD, teleost GSDMEb is activated by inflammatory CASPs. In contrast, teleost GSDMEa is cleaved mainly by apoptotic CASPs. In zebrafish, GSDMEa is cleaved by CASP3 to generate the pyroptotic NT fragment (*Wang et al., 2017*). In turbot (*Scophthalmus maximus*), GSDMEa is bi-directionally regulated by CASP3/7, which activate GSDMEa, and CASP6, which inactivates GSDMEa (*Xu et al., 2022*). In common carp (*Cyprinus carpio haematopterus*), GSDMEa is cleaved by CASP3 and induces pyroptosis (*Zhao et al., 2022*). The complicated scenario of GSDME-mediated pyroptosis signaling in fish is likely due to the reason that, unlike mammals that have multiple GSDM members to induce pyroptosis under different conditions, fish have only GSDME to induce pyroptotic cell death. Regulation by different CASPs may represent a mechanism that enables fish GSDME to execute the orders of different pyroptotic signals.

It is intriguing that, although HsCASP3 and 7 were indistinguishable in proteolysis toward the consensus tetrapeptide, they differed remarkably in the cleavage of HsGSDME and TrGSDME. HsCASP3, but not HsCASP7, cleaved HsGSDME, whereas both HsCASP3 and 7 cleaved TrGSDME at the same site. NT/CT domain swapping between HsGSDME and TrGSDME showed that chimeric GSDMEs containing the TrGSDME-CT domain were cleaved by human as well as pufferfish CASP7 regardless of the source of the NT, whereas chimeric GSDMEs containing the HsGSDME-CT domain were resistant to HsCASP7. These results indicate that the CT domain, which is well accepted to exert an inhibitory effect on the pore-forming NT domain (*Liu et al., 2019*; *Shi et al., 2015*), is the target of HsCASP7 discrimination, and hence determines the cleavability of GSDME by HsCASP7. Recently, Wang and Liu studied the molecular mechanism of human GSDMD recognition by CASP1/4 and showed that GSDMD-CT interacted with CASP1/4 exosites through binding to a hydrophobic pocket, which enhanced the recognition by CASP1/4 and contributed to tetrapeptide sequence-independent cleavage of GSDMD (*Liu et al., 2020*; *Wang et al., 2020b*). Different from GSDMD, we found that for TrGSDME, mutation of either the P4 (D255) or P1 (D258) residue of the consensus motif $^{255}$DAVD$^{258}$ made it resistant to CASP3/7, implying a lack of DAVD-independent cleavage mechanism. Similarly, a previous study observed that human and mouse GSDME harboring cleavage site (P4 or P1) mutation resisted CASP3 cleavage (*Wang et al., 2017*). These results suggest that, compared to GSDMD, GSDME has a distinct enzyme–substrate engagement mode that, as demonstrated in the present study, involves GSDME-CT.

Unlike HsCASP7, which was unable to cleave HsGSDME, TrCASP7 effectively cleaved HsGSDME. The swapping of p20 between HsCASP7 and TrCASP7 revealed that the two catalytic motifs in p20, that is, SHG and QACRG, were not involved in HsCASP7 discrimination on GSDME. SWR and GSWF are known to be responsible for CASP substrate binding (*Chai et al., 2001*; *Riedl et al., 2001*). However, in our study, we found that these two motifs are conserved in human and pufferfish CASP7, and they had no apparent effect on the substrate discrimination of HsCASP7. By contrast, the chimeric

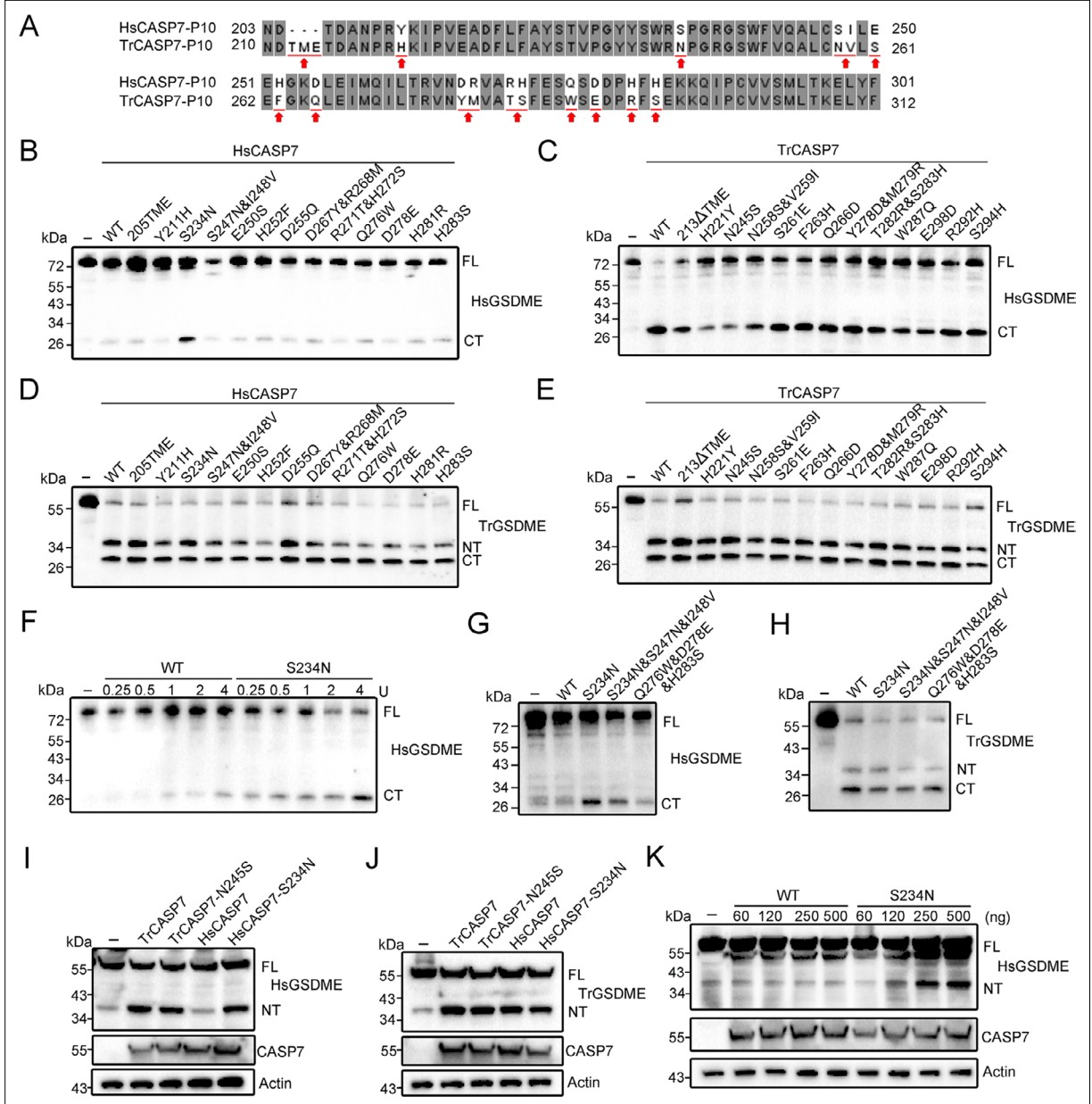

**Figure 5.** Functional importance of the non-conserved residues in the p10 of HsCASP7 and TrCASP7. (**A**) Sequence alignment of the p10 region in HsCASP7 and TrCASP7. The arrows indicate non-conserved residues. (**B, C**) HsGSDME was treated with wild type (WT) or mutant HsCASP7 (**B**) or TrCASP7 (**C**) for 2 hr. The cleavage was determined by immunoblotting with anti-HsGSDME-CT antibody. 205TME, HsCASP7 with TME insertion at position 205; 213ΔTME, TrCASP7 with TME deleted at position 213. (**D, E**) TrGSDME was incubated with wild type (WT) or mutant HsCASP7 (**D**) or TrCASP7 (**E**) for 2 hr. The cleavage was determined by immunoblotting with anti-TrGSDME antibody. (**F**) HsGSDME was incubated with HsCASP7 or the S234N mutant (0.25–4 U) for 2 hr. HsGSDME cleavage was analyzed as above. (**G, H**) HsGSDME (**G**) and TrGSDME (**H**) were treated with wild type or mutant HsCASP7 for 2 hr. The cleavage was determined as above. (**I, J**) TrGSDME (**I**) or HsGSDME (**J**) was co-expressed with wild type or mutant TrCASP7/HsCASP7 for 24 hr. GSDME cleavage, CASP7, and β-actin were determined as above. (**K**) HsGSDME was co-expressed with HsCASP7 or HsCASP7-S234N for 48 hr. Cleavage of GSDME was detected as above. For panels (**B–K**): FL, full length; NT, N-terminal fragment; CT, C-terminal fragment.

The online version of this article includes the following source data and figure supplement(s) for figure 5:

**Source data 1.** Original file for the western blot analysis in *Figure 5*.

**Source data 2.** File containing *Figure 5B–K* and original scans of the relevant western blot analysis with highlighted bands.

**Figure supplement 1.** Cleavage of poly (ADP-ribose) polymerase 1 (PARP1) and gelsolin by HsCASP7.

**Figure supplement 1—source data 1.** Original file for the western blot analysis in *Figure 5—figure supplement 1*.

*Figure 5 continued on next page*

*Figure 5 continued*

**Figure supplement 1—source data 2.** File containing *Figure 5—figure supplement 1* and original scans of the relevant western blot analysis with highlighted bands.

**Figure supplement 2.** Structural analysis of HsCASP1/4/7.

**Figure supplement 3.** GSDME cleavage by CASP7 in cellular system.

**Figure supplement 3—source data 1.** Original file for the western blot analysis in *Figure 5—figure supplement 3*.

**Figure supplement 3—source data 2.** File containing *Figure 5—figure supplement 3* and original scans of the relevant western blot analysis with highlighted bands.

HsCASP7 containing the p10 subunit of TrCASP7 acquired the ability to cleave HsGSDME, indicating an important role of p10. By a series of residue swapping and mutation analyses, we discovered that the residue in the position of 245 (in TrCASP7) or 234 (in HsCASP7) of the p10 subunit was the key that determined CASP7 cleavage of HsGSDME. Sequence analysis revealed that N245 is highly conserved in the CASP7 of teleost, amphibians, reptiles, and birds. In contrast, this Asn residue was changed to Ser in the CASP7 of mammals, especially primates, resulting in divergence of CASP7 from CASP3 in substrate recognition and cleavage. Since, unlike non-mammalian vertebrates, mammals possess multiple pyroptotic GSDM members (GSDMA, B, C, D, and E), the functional divergence of CASP7 from CASP3 may have occurred as a result of specific and non-redundant regulation of programmed cell death mediated by different executor molecules. One of the physiological consequences is that CASP7 frees itself from the GSDME-mediated pathway and thus is able to be engaged in other cellular processes. Another physiological consequence is that GSDME activation is limited to CASP3 cleavage, thus restricting GSDME activity to situations more specific, such as that inducing CASP3 activation. More physiological consequences of CASP3/7 divergence in GSDME activation need to be explored in future studies. With the exception of primates, the conserved Asn is present in a large number of mammals, including mice. Mouse CASP7 was able to cleave HsGSDME, further supporting the importance of the Asn in CASP7 recognition and cleavage. The inability of mouse CASP7 to cleave mouse GSDME is consistent with a previous report that CASP7 deletion had little effect on mouse GSDME-mediated pyroptosis (*Sarhan et al., 2018*). These results suggest mutually independent evolution of GSDME and CASP7 in mice, which likely has distanced GSDME and CASP7 from each other. Similarly, human CASP7 and GSDME may also have undergone independent evolution, which leads to the disengagement of GSDME from the substrate relationship with CASP7.

In conclusion, we identified a teleost GSDME that can be cleaved by fish and human CASP3/7 to trigger pyroptosis. The GSDME-CT domain and the CASP7 p10 subunit are critical in the determination of CASP7 cleavage of GSDME. Within the p10 subunit, a single residue plays a key role in CASP7 substrate recognition and cleavage. Our results reveal the molecular basis of the functional divergence of CASP7 and CASP3, and suggest separate evolutions of CASP7 and GSDME in mammals. These findings add new insights into CASP-regulated GSDME activation in lower and higher vertebrates.

# Materials and methods

**Key resources table**

| Reagent type (species) or resource | Designation | Source or reference | Identifiers | Additional information |
|---|---|---|---|---|
| Gene (*Takifugu rubripes*) | TrGSDME | GenBank | XP_029701077.1 | |
| Strain, strain background (*Escherichia coli*) | Trelief 5a | Tsingke Biological Technology | Cat#TSC01 | |
| Strain, strain background (*E. coli*) | Transetta (DE3) | TransGen | Cat#CD801 | Protein purification |
| Cell line (*Homo sapiens*) | HEK293T | ATCC | Cat#CRL-3216 | |
| Transfected construct (human) | Lipofectamine 3000 | Invitrogen | Cat#L3000075 | |

*Continued on next page*

*Continued*

| Reagent type (species) or resource | Designation | Source or reference | Identifiers | Additional information |
|---|---|---|---|---|
| Antibody | Anti-GSDME (rabbit monoclonal) | Abcam | Cat#ab215191 | WB (1:2000) |
| Antibody | Anti-GSDME (rabbit monoclonal) | Abcam | Cat#ab221843 | WB (1:2000) |
| Antibody | Anti-Flag (mouse polyclonal) | Abclonal | Cat#AE005 | WB (1:5000) |
| Antibody | Anti-Myc (mouse polyclonal) | Abclonal | Cat#AE010 | WB (1:5000) |
| Antibody | Anti-His (mouse polyclonal) | Abclonal | Cat#AE003 | WB (1:5000) |
| Antibody | Anti-mCherry (mouse polyclonal) | Abclonal | Cat#AE002 | WB (1:5000) |
| Antibody | Anti-TrGSDME (mouse polyclonal) | This paper | | Against TrGSDME (1:1000 for WB) |
| Recombinant DNA reagent | pmCherry-N1-TrGSDME variants (plasmid) | This paper | | Transient expression (backbone: pmCherry-N1) |
| Recombinant DNA reagent | pCMV-C-Myc-CASP3/7 (plasmid) | This paper | | Transient expression (backbone: pCMV-C-Myc) |
| Recombinant DNA reagent | pET-30a (+)-TrGSDME (plasmid) | This paper | | Protein purification (backbone: pET-30a) |
| Recombinant DNA reagent | pET-30a (+)-CASPs (plasmid) | This paper | | Protein purification (backbone: pET-30a) |
| Recombinant DNA reagent | pET-28a (+)-SUMO-HsGSDME (plasmid) | This paper | | Protein purification (backbone: pET-28a) |
| Recombinant DNA reagent | pET-28a (+)-SUMO-MmGSDME (plasmid) | This paper | | Protein purification (backbone: pET-28a) |
| Recombinant DNA reagent | pET-28a (+)-SUMO-GSDME chimeras (plasmid) | This paper | | Protein purification (backbone: pET-28a) |
| Sequence-based reagent | Primers | This paper | | *Supplementary file 1e* |
| Peptide, recombinant protein | HsCASP1 | Enzo Life Sciences | Cat#ALX-201-056 | |
| Peptide, recombinant protein | HsCASP2 | Enzo Life Sciences | Cat#ALX-201-057 | |
| Peptide, recombinant protein | HsCASP3 | Enzo Life Sciences | Cat#ALX-201-059 | |
| Peptide, recombinant protein | HsCASP6 | Enzo Life Sciences | Cat#ALX-201-060 | |
| Peptide, recombinant protein | HsCASP7 | Enzo Life Sciences | Cat#ALX-201-060 | |
| Peptide, recombinant protein | HsCASP8 | Enzo Life Sciences | Cat#ALX-201-060 | |
| Peptide, recombinant protein | HsCASP9 | Enzo Life Sciences | Cat#ALX-201-047 | |
| Chemical compound, drug | Ac-YVAD-$\rho$ NA | Enzo Life Sciences | Cat#ALX-260-026 | |
| Chemical compound, drug | Ac-VDQQD-$\rho$ NA | Sigma-Aldrich | Cat#SCP0105 | |
| Chemical compound, drug | Ac-DEVD-$\rho$ NA | Enzo Life Sciences | Cat#ALX-260-033 | |
| Chemical compound, drug | Ac-IETD-$\rho$ NA | Enzo Life Sciences | Cat#ALX-260-045 | |
| Chemical compound, drug | Ac-LEHD-$\rho$ NA | Enzo Life Sciences | Cat#ALX-260-081 | |

*Continued on next page*

*Continued*

| Reagent type (species) or resource | Designation | Source or reference | Identifiers | Additional information |
|---|---|---|---|---|
| Chemical compound, drug | Ac-DAVD-$\rho$ NA | This paper | | Activity comparison of CASP3/7 |
| Chemical compound, drug | Ac-DMPD-$\rho$ NA | This paper | | Activity comparison of CASP3/7 |
| Commercial assay or kit | CytoTox 96 Non-Radioactive Cytotoxicity Assay | Promega | Cat#G1780 | LDH release |
| Commercial assay or kit | Hieff Mut Site-Directed Mutagenesis Kit | Yeasen | Cat#11003ES10 | |
| Commercial assay or kit | FastPure Cell/Tissue Total RNA Isolation Kit V2 | Vazyme | Cat#RC112-01 | |
| Commercial assay or kit | Revert Aid First Strand cDNA Synthesis Kit | Thermo Fisher Scientific | Cat#K1622 | |
| Commercial assay or kit | Endotoxin-free plasmid kit | Sparkjade | Cat#AD0105 | |
| Software, algorithm | Prism software | GraphPad | https://www.graphpad.com/scientific-software/prism/ | |
| Other | Propidium iodide (PI) | Invitrogen | Cat#P1304MP | Cell death examination |
| Other | DiO | Solarbio | Cat#C1038 | Cell death examination |

## Animal, ethics, and cell line

Clinically healthy pufferfish (*T. rubripes*) were obtained from a local fish farm. In the laboratory, the fish were maintained at 19–20°C in aerated seawater as reported previously (*Xu et al., 2022*). For euthanization, the fish were immersed in excess tricaine methane sulfonate (Sigma, St. Louis, MO). The animal experiments were approved by the Ethics Committee of institute of Oceanology, Chinese Academy of Sciences (permit number: MB2012-2). HEK293T cells (ATCC, Rockville, MD) were cultured in DMED medium (Corning, NY) supplemented with 10% fetal bovine serum (ExCell Bio, Shanghai, China) at 37°C in a 5% $CO_2$ incubator. HEK293T cells were validated to be free of mycoplasma contamination before use in this study.

## Sequence analysis

The sequences of 742 CASP3 and 758 CASP7 used in this study were downloaded from NCBI Orthologs. TrCASP7 and HsCASP7 sequences were aligned with Clustal W program (https://www.ebi.ac.uk/jdispatcher/) and visualized with ESPript 3.0 (http://espript.ibcp.fr/ESPript/cgi-bin/ESPript.cgi; *Robert and Gouet, 2014*). Conservation of S/N234 in CASP3/7 was analyzed via Weblogo3 (https://weblogo.threeplusone.com/; *Crooks et al., 2004*), and the sequences of CASP3/7 used are shown in *Supplementary file 1a and b*. The phylogenetic analysis of GSDME was performed as reported previously (*Xu et al., 2022*). The phylogenetic relationship of major mammalian clades was estimated based on the Mammals birth-death node-dated completed trees in VertLife (*Upham et al., 2019*). The phylogenetic tree was subsequently viewed and edited in iTOL (*Letunic and Bork, 2021*). The icons representing phylogeny clades were retrieved from the PhyloPic (http://www.phylopic.org/), with the detailed credentials provided in *Supplementary file 1c*.

## Antibodies and immunoblotting

Monoclonal anti-mouse (ab215191) and anti-human (ab221843) GSDME antibodies were purchased from Abcam (Cambridge, MA). Antibodies against β-actin (AC026), Flag (AE005), Myc (AE010), His (AE003), and mCherry (AE002) were purchased from Abclonal (Wuhang, China). Mouse polyclonal antibody against TrGSDME was prepared as reported previously (*Xu et al., 2022*). Anti-serum (1:1000 dilution) was used for immunoblotting. Immunoblotting was performed as reported previously (*Zhao and Sun, 2022*). Briefly, the samples were fractionated in 12% SDS-PAGE (GenScript, Piscataway, NJ). The proteins were transferred to NC membranes, immunoblotted with appropriate antibodies, and visualized using an ECL kit (Sparkjade Biotechnology Co. Ltd, Shandong, China).

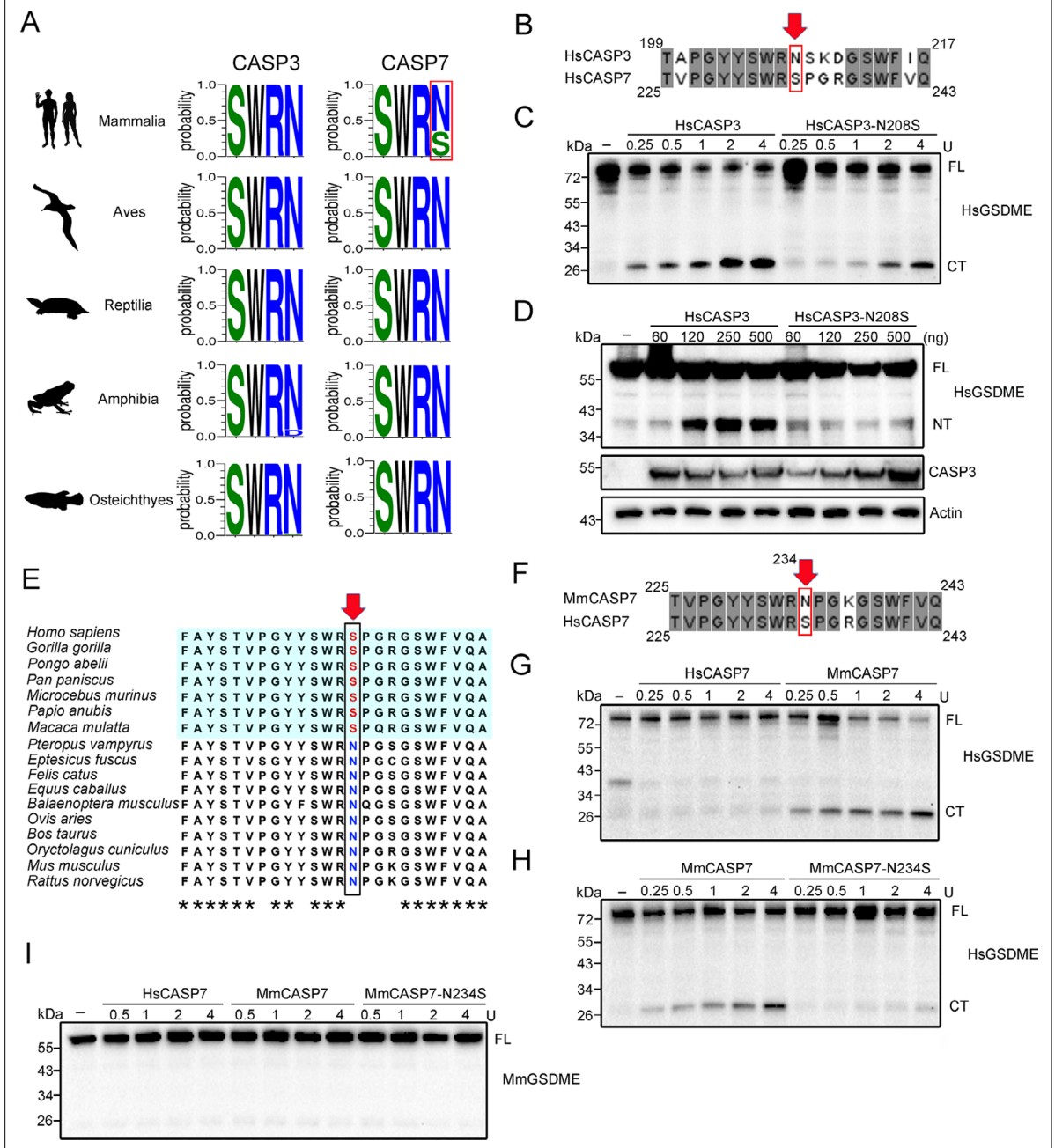

**Figure 6.** Conservation and functional importance of S/N234 in CASP3/7. (**A**) Weblogo analysis of the conservation of S/N234 in CASP3/7 in Mammalia, Aves, Reptilia, Amphibia, and Osteichthyes. (**B**) Sequence alignment of HsCASP3/7. The S/N residues are indicated by red arrow. (**C**) HsGSDME was treated with HsCASP3 or HsCASP3-N208S for 2 hr, and then subjected to immunoblotting with anti-HsGSDME-CT antibody. (**D**) HsGSDME was co-expressed in HEK293T cells with different doses of HsCASP3 or HsCASP3-N208S for 24 hr. The cell lysates were immunoblotted to detect HsGSDME, CASP3, and β-actin. (**E**) Sequence alignment of the S/N234 region in the CASP7 of primate (shaded cyan) and non-primate mammals. The S/N residues are indicated by red arrow. Asterisks indicate identical residues. (**F**) Alignment of human and mouse CASP7 sequences. The S/N234 residues are indicated by arrow. (**G**) HsGSDME was treated with different units of HsCASP7 or MmCASP7 for 2 hr, and the cleavage was assessed by immunoblotting with anti-HsGSDME-CT antibody. (**H**) HsGSDME was incubated with different units of MmCASP7 or its mutant (N234S) for 2 hr, and HsGSDME cleavage was analyzed as above. (**I**) MmCASP7 was treated with HsCASP7, MmCASP7, or MmCASP7-N234S for 2 hr. The cleavage was determined by immunoblotting with anti-MmGSDME-NT antibody. For panels (**C, D, G–I**), FL, full length; NT, N-terminal fragment; CT, C-terminal fragment.

The online version of this article includes the following source data and figure supplement(s) for figure 6:

**Source data 1.** Original file for the western blot analysis in *Figure 6*.

**Source data 2.** File containing *Figure 6C, D–G, and I* and original scans of the relevant western blot analysis with highlighted bands.

*Figure 6 continued on next page*

## Gene cloning and mutagenesis

Total RNA was extracted from pufferfish kidney using the FastPure Cell/Tissue Total RNA Isolation Kit V2 (Vazyme Biotech Co. Ltd, Nanjing, China). cDNA synthesis was performed with Revert Aid First Strand cDNA Synthesis Kit (Thermo Fisher Scientific, Waltham, MA). The coding sequences (CDSs) of pufferfish GSDME (accession number XP_029701077.1) and CASP3/7 were amplified by PCR. The CDSs of mouse GSDME and CASP3/7 were synthesized by Sangon Biotech (Shanghai, China). Site-directed mutagenesis was performed using Hieff Mut Site-Directed Mutagenesis Kit (Yeasen, Shanghai, China). Truncates of TrGSDME were created as reported previously (*Xu et al., 2022*) and subcloned into pmCherry-N1 (Clontech, Mountain View, CA). For the construction of GSDME and CASP7 chimera, the CDSs of GSDME NT/CT and CASP7 p10/p20 were obtained by PCR and ligated. The constructs of the truncate and chimeric proteins are shown in *Supplementary file 1d*. All sequences/constructs were verified by sequencing analysis. The primers used are listed in *Supplementary file 1e*.

## Plasmids and transient expression

For expression in mammalian cell expression, the CDSs of GSDME (pufferfish, human, and mouse) were subcloned into pCS2–3×Flag (*Wang et al., 2017*) or pmCherry-N1 and CDSs encoding CASP (pufferfish, human, and mouse) were subcloned into p-CMV-Myc (Clontech) or pCS2-Myc (*Wang et al., 2017*). All plasmids were prepared with endotoxin-free plasmid kit (Sparkjade Biotechnology Co. Ltd). For transient expression, HEK293T cells were cultured in 96- or 24-well plates (Corning) overnight, and then transfected with 100 ng/well (96-well plates) or 500 ng/well (24-well plates) indicated plasmids using Lipofectamine 3000 (Invitrogen, USA) for 24 hr or as specified. For GSDME processing, the plasmids of GSDME and CASP were co-transfected in HEK293T cells as described above and cultured continuously for 24 hr or as specified. Cells were harvested and lysed with RIPA Lysis Buffer (Beyotime, Shanghai, China) for immunoblotting. The primers used are shown in *Supplementary file 1e*.

## Protein purification

Recombinant GSDMEs and CASPs are all soluble and were purified as described previously (*Jiang et al., 2020*; *Xu et al., 2022*). Briefly, the CDSs of TrGSDME and CASP variants were each cloned into pET30a (+), and the CDSs of HsGSDME, MmGSDME, and chimeric GSDME were each cloned into pET28a-SUMO. The recombinant plasmids were introduced into *Escherichia coli* Transetta (DE3) (TransGen, Beijing, China) by transformation. The transformants were cultured in Luria broth (LB) at 37°C until logarithmic growth phase. Isopropyl-β-D-thiogalactopyranoside (0.3 mM) was added to the medium, followed by incubation at 16°C for 20 hr. Bacteria were harvested and lysed, and the supernatant was collected for protein purification with Ni-NTA columns (GE Healthcare, Uppsala, Sweden). The proteins were dialyzed with PBS at 4°C and concentrated.

## CASP activity

To measure their proteolytic activity, recombinant CASPs were incubated separately with various colorimetric substrates at 37°C for 2 hr as described previously (*Jiang et al., 2020*; *Xu et al., 2022*), and then monitored for released $\rho$NA at OD405. To compare their substrate preference, TrCASP3/7 and HsCASP3/7 were incubated with Ac-DAVD-$\rho$NA or Ac-DMPD-$\rho$NA (Science Peptide Biological Technology Co., Ltd, Shanghai, China) at 37°C for 30 min, and the released $\rho$NA was measured at OD405. The CASP activity in cells transfected TrCASP3/7 was determined as described previously (*Xu et al., 2022*).

## GSDME cleavage by CASPs

GSDME cleavage assay was performed as described previously (*Jiang et al., 2020*; *Xu et al., 2022*). Briefly, recombinant GSDME was incubated with 1 U of recombinant HsCASP1,2, 3, 6, 7, 8, and 9 (Enzo Life Sciences, Villeurbanne, France) at 37°C for 2 hr in the reaction buffer (50 mM HEPES [pH 7.5], 3 mM EDTA, 150 mM NaCl, 0.005% [v/v] Tween 20, and 10 mM DTT). TrGSDME, HsGSDME, or their chimeras were treated with 1 U of TrCASP3/7, HsCASP3/7, or their chimeras for 2 hr as above in reaction buffer. After incubation, the samples were boiled in 5× Loading Buffer (GenScript) and subjected to SDS-PAGE or immunoblotting with indicated antibody.

## Cell death examination

Cell death examination by microscopy was performed as reported previously (*Jiang et al., 2020*; *Xu et al., 2022*). Briefly, HEK293T cells were plated into 35 mm glass-bottom culture dishes (Nest Biotechnology, Wuxi, China) at about 60% confluency and subjected to the indicated treatment for 24 hr. To examine cell death morphology, propidium iodide (PI) (Invitrogen, Carlsbad, CA) and DiO (Solarbio, Beijing, China) were added to the culture medium, and the cells were observed with a Carl Zeiss LSM 710 confocal microscope (Carl Zeiss, Jena, Germany). To video the cell death process, the cells were transfected with 1 μg pmCherry-N1 vector expressing TrGSDME-NT for 16 hr, and cell death was recorded using the above microscope. Cell death measured by LDH assay was performed as reported previously (*Xu et al., 2022*).

## Statistical analysis

Student's *t*-test and ANOVA were used for comparisons between groups. Statistical analysis was performed with GraphPad Prism 7 software. Statistical significance was defined as $p < 0.05$.

## Acknowledgements

We thank Prof. Feng Shao (National institute of Biological Sciences, Beijing, China) for providing human CASP3/7 and GSDME expression vectors. This work was supported by grants from the Science & Technology Innovation Project of Laoshan Laboratory (LSKJ202203000), the Youth Innovation Promotion Association CAS (2021204), the National Natural Science Foundation of China (41876175), and the Taishan Scholar Program of Shandong Province (2018 and 2021).

## Additional information

### Funding

| Funder | Grant reference number | Author |
|---|---|---|
| Science and Technology Innovation Project of Laoshan Laboratory | LSKJ202203000 | Li Sun |
| Youth Innovation Promotion Association CAS | 2021204 | Shuai Jiang |
| National Natural Science Foundation of China | 41876175 | Shuai Jiang |
| Taishan Scholar Program of Shandong Province | 2018 and 2021 | Li Sun Shuai Jiang |

The funders had no role in study design, data collection and interpretation, or the decision to submit the work for publication.

### Author contributions

Hang Xu, Data curation, Software, Formal analysis, Investigation, Visualization, Methodology, Writing - original draft; Zihao Yuan, Kunpeng Qin, Investigation; Shuai Jiang, Conceptualization, Supervision,

Writing - review and editing; Li Sun, Conceptualization, Supervision, Funding acquisition, Writing - review and editing

**Author ORCIDs**
Hang Xu [ID] http://orcid.org/0000-0002-1378-5922
Shuai Jiang [ID] http://orcid.org/0000-0003-3027-083X
Li Sun [ID] http://orcid.org/0000-0001-7183-7148

**Ethics**
The animal experiments were approved by the Ethics Committee of institute of Oceanology, Chinese Academy of Sciences (permit number: MB2012-2).

Reviewer #1 (Public Review): https://doi.org/10.7554/eLife.89974.3.sa1
Reviewer #2 (Public Review): https://doi.org/10.7554/eLife.89974.3.sa2
Author Response https://doi.org/10.7554/eLife.89974.3.sa3

## Additional files

### Supplementary files
• Supplementary file 1. Supplementary tables. (**a**) The CASP3 sequences used for WebLogo analysis in this study. (**b**) The CASP7 sequences used for WebLogo analysis in this study. (**c**) The credits for the pictures used in this study. (**d**) The constructs of the truncate and chimeric proteins. (**e**) The primers used in this study.

• MDAR checklist

### Data availability
All data generated or analysed during this study are included in the manuscript and supporting files.

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
