## [Editor Report · eLife assessment]

This **important** study elucidates the molecular divergence of caspase 3 and 7 in the vertebrate lineage. **Convincing** biochemical and mutational data provide evidence that in humans, caspase 7 has lost the ability to cleave gasdermin E due to changes in a key residue, S234. The diversification and specialization of gasdermins such as gasdermin E in humans compared to early vertebrates such as teleosts may enable each human gasdermin molecule to have more restricted and tightly regulated physiological functions in different cell death pathways.

---

## [Referee Report · Reviewer #1 (Public Review)]

Summary

In this study, Xu et al. provide insights into the substrate divergence of caspase 3 and 7 (CASP3 and CASP7) for gasdermin E (GSDME) cleavage and activation during evolution in vertebrates. Using a diverse set of biochemical assays, domain swapping, site-directed mutagenesis, and bioinformatics tools, the authors demonstrate that the human GSDME C-terminal region and the S234 residue of human CASP7 are the key determinants that impede the cleavage of human GSDME by human CASP7. Their findings suggest that mutations affecting the function of caspases have enabled the functional divergence of distinct caspase family members to specialize in controlling complicated cellular functions in mammals.

Strengths

The authors made an important contribution to the field by demonstrating how human CASP7 has functionally diverged to lose the ability to cleave GSDME and showing that reverse-mutations in CASP7 can restore GSDME cleavage. The use of multiple methods to support their conclusions strengthens the authors' findings. The unbiased mutagenesis screen performed to identify S234 in huCASP7 as the determinant of its GSDME cleavability is also a strength.

Weaknesses

While the authors employed a comprehensive experimental setup to investigate the CASP7-mediated GSDME cleavage across evolution, future studies will be required to fully understand the physiological implications of this evolutionary divergence.

---

## [Referee Report · Reviewer #2 (Public Review)]

The authors wanted to address the differential processing of GSDME by caspase 3 and 7, finding that while in humans GSDME is only processed by CASP3, Takifugu GSDME, and other mammalian can be processed by CASP3 and 7. This is due to a change in a residue in the human CAPS7 active site that abrogates GSDME cleavage. This phenomenon is present in humans and other primates, but not in other mammals such as cats or rodents. This study sheds light on the evolutionary changes inside CASP7, using sequences from different species. Although the study is somehow interesting and elegantly provides strong evidence of this observation, it lacks the physiological relevance of this finding, i.e. on human side, mouse side, and fish what are the consequences of CASP3/7 vs CASP3 cleavage of GSDME.

Fish also present a duplication of GSDME gene and Takifugu present GSDMEa and GSDMEb. It is not clear in the whole study if when referring to TrGSDME is the a or b. This should be stated in the text and discussed in the differential function of both GSDME in fish physiology (i.e. PMIDs: 34252476, 32111733 or 36685536).

---

## [Author Response]

The following is the authors’ response to the original reviews.

**eLife assessment**
This important study elucidates the molecular divergence of caspase 3 and 7 in the vertebrate lineage. Convincing biochemical and mutational data provide evidence that in humans, caspase 7 has lost the ability to cleave gasdermin E due to changes in a key residue, S234. However, the physiological relevance of the findings is incomplete and requires further experimental work.
**Public Reviews:**

**Reviewer #1 (Public Review):**
SummaryIn this study, Xu et al. provide insights into the substrate divergence of CASP3 and CASP7 for GSDME cleavage and activation during vertebrate evolution vertebrates. Using biochemical assays, domain swapping, site-directed mutagenesis, and bioinformatics tools, the authors demonstrate that the human GSDME C-terminal region and the S234 residue of human CASP7 are the key determinants that impede the cleavage of human GSDME by human CASP7.StrengthsThe authors made an important contribution to the field by demonstrating how human CASP7 has functionally diverged to lose the ability to cleave GSDME and showing that reverse-mutations in CASP7 can restore GSDME cleavage. The use of multiple methods to support their conclusions strengthens the authors' findings. The unbiased mutagenesis screen performed to identify S234 in huCASP7 as the determinant of its GSDME cleavability is also a strength.WeaknessesWhile the authors utilized an in-depth experimental setup to understand the CASP7-mediated GSDME cleavage across evolution, the physiological relevance of their findings are not assessed in detail. Additional methodology information should also be provided.Specific recommendations for the authors(1) The authors should expand their evaluation of the physiological relevance by assessing GSDME cleavage by the human CASP7 S234N mutant in response to triggers such as etoposide or VSV, which are known to induce CASP3 to cleave GSDME (PMID: 28045099). The authors could also test whether the human CASP7 S234N mutation affects substrate preference beyond human GSDME by testing cleavage of mouse GSDME and other CASP3 and CASP7 substrates in this mutant.

(1) The physiological relevance was discussed in the revised manuscript (lines 328-340). Our study revealed the molecular mechanism underlying the divergence of CASP3- and CASP7-mediated GSDME activation in vertebrate. One of the physiological consequences is that in humans, CASP7 no longer directly participates in GSDME-mediated cell death, which enables CASP7 to be engaged in other cellular processes. Another physiological consequence is that GSDME activation is limited to CASP3 cleavage, thus restricting GSDME activity to situations more specific, such as that inducing CASP3 activation. The divergence and specialization of the physiological functions of different CASPs are consistent with and possibly conducive to the development of refined regulations of the sophisticated human GSDM pathways, which are executed by multiple GSDM members (A , B, C, D, and E), rather than by GSDME solely in teleost, such as Takifugu. More physiological consequences of CASP3/7 divergence in GSDME activation need to be explored in future studies.

With respect to the reviewer’s suggestion of assessing GSDME cleavage by the human CASP7 S234N mutant in response to triggers such as etoposide or VSV: (i) CASP7 S234N is a creation of our study, not a natural human product, hence its response to CASP7 triggers cannot happen under normal physiological conditions except in the case of application, such as medical application, which is not the aim of our study. (ii) CASP3/7 activators (such as raptinal) induced robust activation of the endogenous CASP3 (Heimer et al., Cell Death Dis. 2019;10:556) and CASP7 (Author response image 1, below) in human cells. Since CASP3 is the natural activator of GSDME, the presence of the triggers inevitably activates GSDME via CASP3. Hence, under this condition, it will be difficult to examine the effect of CASP7 S234N.

**Author response image 1. sa3fig1:** HsCASP7 activation by raptinal. HEK293T cells were transfected with the empty vector (-), or the vector expressing HsCASP7 or HsCASP7-S234N for 24 h. The cells were then treated with or without (control) 5 μM raptinal for 4 h. The cells were lysed, and the lysates were blotted with anti-CASP7 antibody.

(2) As suggested by the reviewer, the cleavage of other CASP7 substrates, i.e., poly (ADP-ribose) polymerase 1 (PARP1) and gelsolin, by HsCASP7 and S234N mutant was determined. The results showed that HsCASP7 and HsCASP7-S234N exhibited similar cleavage capacities. Figure 5-figure supplement 1 and lines 212-214.

(2) It would also be interesting to examine the GSDME structure in different species to gain insight into the nature of mouse GSDME, which cannot be cleaved by either mouse or human CASP7.

Because the three-dimensional structure of GSDME is not solved, we are unable to explore the structural mechanism underlying the GSDME cleavage by caspase. Since our results showed that the C-terminal domain was essential for caspase-mediated cleavage of GSDME, it is likely that the C-terminal domain of mouse GSDME may possess some specific features that render it to resist mouse and human CASP7.

(3) The evolutionary analysis does not explain why mammalian CASP7 evolved independently to acquire an amino acid change (N234 to S234) in the substrate-binding motif. Since it is difficult to experimentally identify why a functional divergence occurs, it would be beneficial for the authors to speculate on how CASP7 may have acquired functional divergence in mammals; potentially this occurred because of functional redundancies in cell death pathways, for example.

According to the reviewer’s suggestion, a speculation was added. Lines 328-340.

(4) For the recombinant proteins produced for these analyses, it would be helpful to know whether size-exclusion chromatography was used to purify these proteins and whether these purified proteins are soluble. Additionally, the SDS-PAGE in Figure S1B and C show multiple bands for recombinant mutants of TrCASP7 and HsCASP7. Performing protein ID to confirm that the detected bands belong to the respective proteins would be beneficial.

The recombinant proteins in this study are soluble and purified by Ni-NTA affinity chromatography. Size-exclusion chromatography was not used in protein purification.

For the SDS-PAGE in Figure 4-figure supplement 1B and C (Figure S1B and C in the previous submission), the multiple bands are most likely due to the activation cleavage of the TrCASP7 and HsCASP7 variants, which can result in multiple bands, including p10 and p20. According to the reviewer’s suggestion, the cleaved p10 was verified by immunoblotting. Figure 4-figure supplement 1B and C.

(5) For Figures 3C and 4A, it would be helpful to mention what parameters or PDB files were used to attribute these secondary structural features to the proteins. In particular, in Figure 3C, residues 261-266 are displayed as a β-strand; however, the well-known α-model represents this region as a loop. Providing the parameters used for these callouts could explain this difference.

For Figure 3C, in the revised manuscript, we used the structure of mouse GSDMA3 (PDB: 5b5r) for the structural analysis of HsGSDME. As indicated by the reviewer, the region of 261-266 is a loop. The description was revised in lines 172 and 174, Figure 3C and Figure 3C legend.

For Figure 4A, the alignment of CASP7 was constructed by using Esprit (https://espript.ibcp.fr/ESPript/cgi-bin/ESPript.cgi) with human CASP7 (PDB:1k86) as the template. The description was revised in the Figure legend.

(6) Were divergent sequences selected for the sequence alignment analyses (particularly in Figure 6A)? The selection of sequences can directly influence the outcome of the amino acid residues in each position, and using diverse sequences can reduce the impact of the number of sequences on the LOGO in each phylogenetic group.

In Figure 6A, the sequences were selected without bias. For Mammalia, 45 CASP3 and 43 CASP7 were selected; for Aves, 41 CASP3 and 52 CASP7 were selected; for Reptilia, 31CASP3 and 39 CASP7 were selected; for Amphibia, 11 CASP3 and 12 CASP7 were selected; for Osteichthyes, 40 CASP3 and 43 CASP7 were selected. The sequence information was shown in Table 1 and Table 2.

(7) For clarity, it would help if the authors provided additional rationale for the selection of residues for mutagenesis, such as selecting Q276, D278, and H283 as exosite residues, when the CASP7 PDB structures (4jr2, 3ibf, and 1k86) suggest that these residues are enriched with loop elements rather than the β sheets expected to facilitate substrate recognition in exosites for caspases (PMID: 32109412). It is possible that the inability to form β-sheets around these positions might indicate the absence of an exosite in CASP7, which further supports the functional effect of the exosite mutations performed.

According to the suggestion, the rationale for the selection of residues for mutagenesis was added (lines 216-222). Unlike the exosite in HsCASP1/4, which is located in a β sheet, the Q276, D278, and H283 of HsCASP7 are located in a loop region (Figure 5-figure supplement 2), which may explain the mutation results and the absence of an exosite in HsCASP7 as suggested by the reviewer.

**Reviewer #2 (Public Review):**
The authors wanted to address the differential processing of GSDME by caspase 3 and 7, finding that while in humans GSDME is only processed by CASP3, Takifugu GSDME, and other mammalian can be processed by CASP3 and 7. This is due to a change in a residue in the human CAPS7 active site that abrogates GSDME cleavage. This phenomenon is present in humans and other primates, but not in other mammals such as cats or rodents. This study sheds light on the evolutionary changes inside CASP7, using sequences from different species. Although the study is somehow interesting and elegantly provides strong evidence of this observation, it lacks the physiological relevance of this finding, i.e. on human side, mouse side, and fish what are the consequences of CASP3/7 vs CASP3 cleavage of GSDME.

Our study revealed the molecular mechanism underlying the divergence of CASP3- and CASP7-mediated GSDME activation in vertebrate. One of the physiological consequences is that in humans, CASP7 no longer directly participates in GSDME-mediated cell death, which enables CASP7 to be engaged in other cellular processes. Another physiological consequence is that GSDME activation is limited to CASP3 cleavage, thus restricting GSDME activity to situations more specific, such as that inducing CASP3 activation. The divergence and specialization of the physiological functions of different CASPs are consistent with and possibly conducive to the development of refined regulations of the sophisticated human GSDM pathways, which are executed by multiple GSDM members (A , B, C, D, and E), rather than by GSDME solely in teleost, such as Takifugu. More physiological consequences of CASP3/7 divergence in GSDME activation need to be explored in future studies. Lines 328-340.

Fish also present a duplication of GSDME gene and Takifugu present GSDMEa and GSDMEb. It is not clear in the whole study if when referring to TrGSDME is the a or b. This should be stated in the text and discussed in the differential function of both GSDME in fish physiology (i.e. PMIDs: 34252476, 32111733 or 36685536).

The TrGSDME used in this study belongs to the GSDMEa lineage of teleost GSDME. The relevant information was added. Figure 1-figure supplement 1 and lines 119, 271, 274-276, 287 and 288.

**Recommendations for the authors:**

**Reviewer #1 (Recommendations For The Authors):**
(1) For the chimeric and truncated constructs, such as HsNT-TrCT, TrNT-HsCT, Hsp20-Trp10, Trp20-Hsp10, etc., the authors should provide a table denoting which amino acids were taken from each protein to create the fusion or truncation.

According to the reviewer’s suggestion, the information of the truncate/chimeric proteins was provided in Table 4.

(2) Both reviewers agree that functional physiological experiments are needed to increase the significance of the work. Specifically, the physiological relevance of these findings can be assessed by using western blotting to monitor GSDME cleavage by the human CASP7 S234N mutant compared with wild type CASP7 in response to triggers such as etoposide or VSV, which are known to induce CASP3 to cleave GSDME (PMID: 28045099).Additionally, the authors can assess cell death in HEK293 cells, HEK293 cells transfected with TrGSDME, HEK293 cells expressing TrCASP3/7 plus TrGSDME, and TrCASP3/7 plus the D255R/D258A mutant. These cells can be stimulated, and pyroptosis can be assessed by using ELISA to measure the release of the cytoplasmic enzyme LDH as well as IL-1β and IL-18, and the percentage of cell death (PI+ positive cells) may also be assessed.

(1) With respect to the physiological relevance, please see the above reply to Reviewer 1’s comment of “Specific recommendations for the authors, 1”.

(2) As shown in our results (Fig. 2), co-expression of TrCASP3/7 and TrGSDME in HEK293T cells induced robust cell death without the need of any stimulation, as evidenced by LDH release and TrGSDME cleavage. In the revised manuscript, similar experiments were performed as suggested, and cell death was assessed by Sytox Green staining (Figure 2-figure supplement 3A and B) and immunoblot to detect the cleavage of both wild type and mutant TrGSDME (Figure 2-figure supplement 3C). The results confirmed the results of Figure 2.

**Reviewer #2 (Recommendations For The Authors):**
Abstract:Although the authors try to summarize the principal results of this study, please rewrite the abstract section to make it easier to follow and to empathise the implications of their results.

We have modified the Abstract as suggested by the reviewer.

Introduction:The authors do not mention anything about the implication of the inflammasome activation to get pyroptosis by GSDM cleave by inflammatory caspases. Please consider including this in the introduction section as they do in the discussion section.

The introduction was modified according to the reviewer’s suggestion. Lines 58-61.

From the results section the authors name the human GSDM as HsGSDM and the human CASP as HsCASP, maybe the author could use the same nomenclature in the introduction section. The same for the fish GSDM (Tr) and CASP.

According to the reviewer’s suggestion, the same nomenclature was used in the introduction.

Line 39. Remove the word necrotic.

“necrotic” was removed .

Line 42. Change channels by pores. In the manuscript, change channels by pores overall.

“channels” was replaced by “pores”.

Line 42: Include that: by these pores can be released the proinflammatory cytokines and if these pores are not solved then pyroptosis occurs. Please rephrase this statement.

According to the reviewer's suggestion, the sentence was rephrased. Lines 46-48.

Line 45. GSDMF is not an approved gene name, its official nomenclature is PJVK (Uniprot Q0ZLH3). Please use PJVK instead GSDMF.

GSDMF was changed to PJVK.

Line 103: Can the authors explain better the molecular determinant?

The sentence was revised, line 109.

Results:Line 110: Reference for this statement.The reference for this statement was added in line 116.Figure 1A, B: Concentration or units used of HsCASP?

The unit (1 U) of HsCASPs was added to the figure legend (line 661).

Line 113: Add Hs or Tr after CASP would be helpful to follow the story.

“CASP” was changed to “HsCASP”.

Fig 1D: Why the authors do not use the DMPD tetrapeptide (HsGSDME CASP3 cut site) in this assay? Comparing with the data obtained in Fig 3B the TrCASP3 activity is going to be very closer to that obtained for VEID o VDQQD in the CASP3 panel.

The purpose of Figure 1D was to determine the cleavage preference of TrCASPs. For this purpose, a series of commercially available CASP substrates were used, including DEVD, which is commonly used as a testing substrate for CASP3. Figure 3B was to compare the cleavage of HsCASP3/7 and TrCASP3/7 specifically against the motifs from TrGSDME (DAVD) and HsGSDME (DMPD).

Figure 1D and Figure 3B are different experiments and were performed under different conditions. In Figure 1D, CASP3 was incubated with the commercial substrates at 37 ℃ for 2 h, while in Figure 3B, CASP3/7 were incubated with non-commercial DAVD (motif from TrGSDME) and DMPD (motif from HsGSDME) at 37 ℃ for 30 min. More experimental details were added to Materials and Methods, lines 443 and 447.

Fig 1H: What is the concentration used of the inhibitors?

The concentration (20 μM) was added to the figure legend (line 669).

Does the Hs CASP3/7 fail to cleave the TrGSDME mutants (D255R and D258A)? the authors do not show this result so they cannot assume that HsCASP3/7 cleave that sequence (although this is to be expected).

The result of HsCASP3/7 cleavage of the TrGSDME mutants was added as Figure 1-figure supplement 2 and described in Results, line 133.

Line 132-133: Can the author specify where is placed the mCherry tag? In the N terminal or C terminal portion of the different engineered proteins?

The mCherry tag is attached to the C-terminus. Figure 2 legend (line 676).

Fig 2A: Although is quite clear, a column histogram showing the quantification is going to be helpful.

The expression of TrGSDME-FL, -NT and -CT was determined by Western blot, and the result was added as Figure 2-figure supplement 1.

Fig 2A, B, C: After how many hours of expression are the pictures taken? Can the authors show a Western blot showing that the expression of the different constructions is similar?

The time was added to Figure 2 legend and Materials and Methods (line 466). The expression of TrGSDME-FL, -NT and -CT was determined by Western blot, and the result was added as Figure 2-figure supplement 1.

Fig 2C: Another helpful assay can be to measure the YO-PRO or another small dye internalization, to complete the LDH data.

According the reviewer’s suggestion, in addition to LDH release, Sytox Green was also used to detect cell death. The result was added as Figure 2-figure supplement 2 and described in Results, line 146.

Fig 2C: In the figure y axe change LHD by LDH.

The word was corrected.

Fig 2D: Change HKE293T by HEK293T in the caption.

The word was corrected.

Fig 2G: Please add the concentration used with the two plasmids co-transfection. A Western blot showing CASP3/7 expression vs TrGSDME is missing. Is that assay after 24h? please specify better the methodology.

The concentration of plasmid used in co-transfection and the time post transfection were added to the Materials and Methods (lines 422 and 424). In addition, the expression of CASP3/7 was added to Figure 2I.

Fig 2 J, K: Change HKE293T by HEK293T in the figure caption. The concentration of the caspase inhibitors is missing. Depending on the concentration used, these inhibitors used could provoke toxicity on the cells by themselves.

The word was corrected in the figure caption. The inhibitor concentration (10 μM) was added to the figure legend (line 690).

Line 151: TrCASP3/7 instead of CASP3/7

CASP3/7 was changed to TrCASP3/7.

Fig 3A, 3B: Please add the units used of the HsCASP

The unit was added to the figure legends (lines 697).

Fig 3A: Can the authors add the SDS-PAGE to see the Nt terminal portion as has been done in Fig 1A? Maybe in a supplementary figure.

The SDS-PAGE was added as Figure 3-figure supplement 1.

Fig 3B: If the authors could add some data about the caspase activity using any other CASP such as CASP2, CASP1 to compare the activity data with CASP3 and CASP7 would be helpful.

The proteolytic activity of TrCASP1 was provided as Figure 3-figure supplement 2.

Fig 3C: To state this (Line 160), the authors should use another prediction software to reach a consensus with the sequences of the first analysis. In fact, what happens when GSDME is modelled 3-dimensionally by comparing it to crystalized structures such as mouse GSDMA? If the authors add an arrow indicating where the Nt terminal portion ends and where Ct portion begins would make the figure clearer.

According to the suggestions of both reviewers, in the revised manuscript, we used mouse GSDMA3 (PDB: 5b5r) for the structural analysis of HsGSDME, which showed that the 261-266 region of HsGSDME was a loop. As a result, Figure 3C was revised. Relevant change in Results: lines 172 and 174.

As suggested by the reviewer, we modelled the three-dimensional structure of HsGSDME by using SWISS-MODEL with mouse GSDMA3 as the template (Author response image 2, below).

**Author response image 2. sa3fig2:** The three-dimensional structure model of HsGSDME. (A) The structure of HsGSDME was modeled by using mouse GSDMA3 (MmGSDMA3) as the template. The N-terminal domain (1-246 aa) and the C-terminal domain (279-468 aa) of HsGSDME are shown in red and blue, respectively. (B) The superposed structure of HsGSDME (cyan) and MmGSDMA3 (purple).

Fig 3F: if this is an immunoblotting why NT can be seen? In other Western blots only the CT is detected, why? The use of the TrGSDME mouse polyclonal needs more details (is a purify Ab, was produced for this study, what are the dilution used...)

Since the anti-TrGSDME antibody was generated using the full-length TrGSDME, it reacted with both the N-terminal and the C-terminal fragments of TrGSDME in Figure 3F. In Figure 3G, the GSDME chimera contained only TrGSDME-CT, so only the CT fragment was detected by anti-TrGSDME antibody. More information on antibody preparation and immunoblot was added to “Materials and Methods” (lines 390 and 391).

Fig 4B: Can the authors show in which amino acid the p20 finish for each CASP? (Similarly, as they have done in panel 3E)

Fig 4B was revised as suggested.

Fig 5F: With 4 units of WT CASP7 the authors show a HsGSDME Ct in the same proportion than when the S234N mutant is used (at lower concentrations). How do the authors explain this?

The result showed that the cleavage by 4U of HsCASP7 was comparable to the cleavage by 0.25U of HsCASP7-S234N, indicating that S234 mutation increased the cleavage ability of HsCASP7 by 16 folds.

Line 203: Can the authors show an alignment between this region of casp1/4 and 7? Maybe in supplementary figures.

As reported by Wang et. al (PMID: 32109412), the βIII/βIII’ sheet of CASP1/4 forms the exosite critical for GSDMD recognition. The structural comparison among HsCASP1/4/7 and the sequence alignment of HsCASP1/4 βIII/βIII’ region with its corresponding region in HsCASP7 were added as Figure 5-figure supplement 2.

Line 205: A mutation including S234N with the exosite mutations (S234+Q276W+D278E+H283S) is required to support this statement.

The sentence of “suggesting that, unlike human GSDMD, HsGSDME cleavage by CASPs probably did not involve exosite interaction” was deleted in the revised manuscript.

Fig 5I, 5J: which is the amount of HsGSDME and TrGSDME? I would place these figures in supplementary material.

The protein expression of TrGSDME/HsGSDME was shown in the figure. Fig 5I and 5J were moved to Figure 5-figure supplement 3.

Line 218: I would specify that this importance is in HUMAN CASP7 to cleavage Human GSDME.

“CASP7” and “GSDME” were changed to “HsCASP7” and “HsGSDME”, respectively.

Fig 6C: 4 units is the amount of S234N mutant needed to see an optimal HsGSDME cleavage in Fig 5F.

In Figure 6C, the cleavage efficacy of HsCASP3-N208S was apparently decreased compared to that of HsCASP3, and 4U of HsCASP3-N208S was roughly equivalent to 1U of HsCASP3 in cleavage efficacy. In Figure 5F, cleavage by 4U of HsCASP7 was comparable to the cleavage by 0.25U of HsCASP7-S234N. Together, these results confirmed the critical role of S234/N208 in HsCASP3/7 cleavage of HsGSDM.

Fig 6I: Could be the fact that the mouse GSDME has a longer Ct than human GSDME affect the interaction with CASP7? Less accessible to the cut site? Needs a positive control of mouse GSDME with mouse Caspase 3.

Although mouse GSDME (MmGSDME) (512 aa) is larger than HsGSDME (496 aa), the length of the C-terminal domain of MmGSDME (186 aa) is comparable to that of HsGSDME (190 aa).

**Author response image 3. sa3fig3:** Conserved domain analysis of mouse (upper) and human (lower) GSDME.

As suggested by the reviewer, the cleavage of MmGSDME by mouse caspase-3 (MmCASP3) was added as Figure 6-figure supplement 2 and described in Results, lines 258.

Material and Methods:-Overall, concentrations or amounts used in this study regarding the active enzyme or plasmids used are missing and need to be added.

The missing concentrations of the enzymes and plasmids were added in Material and Methods (lines 421, 453, 457, and 470) or figure legends (Figure 1 and 3).

-It would be helpful if the authors label in the immunoblotting panels what is the GSDME that they are using. (Hs GSDME FL...).

As suggested, the labels were added to Figures 1A ,1B, and 3.

-Add the units of enzyme used.

The units of enzyme were added to figure legends (Figure 1A, 3A, 3D, and 3F) or Material and Methods (lines 453 and 457).

The GSDME sequence obtained for Takifugu after amplification of the RNA extracted should be shown and specified (GSDMEa or GSDMEb). From which tissue was the RNA extracted?

The details were added to Materials and Methods (lines 398 and 402).